# Benchmarking Estimators for Natural Experiments: A Novel Dataset and a Doubly Robust Algorithm

**R. Teal Witter**
New York University
`rtealwitter@nyu.edu`

**Christopher Musco**
New York University
`cmusco@nyu.edu`

## Abstract

Estimating the effect of treatments from natural experiments, where treatments are pre-assigned, is an important and well-studied problem. We introduce a novel natural experiment dataset obtained from an early childhood literacy nonprofit. Surprisingly, applying over 20 established estimators to the dataset produces inconsistent results in evaluating the nonprofit's efficacy. To address this, we create a benchmark to evaluate estimator accuracy using synthetic outcomes, whose design was guided by domain experts. The benchmark extensively explores performance as real world conditions like sample size, treatment correlation, and propensity score accuracy vary. Based on our benchmark, we observe that the class of *doubly robust* treatment effect estimators, which are based on simple and intuitive regression adjustment, generally outperform other more complicated estimators by orders of magnitude. To better support our theoretical understanding of doubly robust estimators, we derive a closed form expression for the variance of any such estimator that uses dataset splitting to obtain an unbiased estimate. This expression motivates the design of a new doubly robust estimator that uses a novel loss function when fitting functions for regression adjustment. We release the dataset and benchmark in a Python package; the package is built in a modular way to facilitate new datasets and estimators.[1]

## 1 Introduction

In this work we consider the problem of *treatment effect estimation*, which is ubiquitous in the sciences, social sciences, economics, and a number of other fields [MLDF15, JT16, AI16]. We focus on the challenging setting where we do not have access to data from a randomized control trial. Instead, treatment effect must be estimated from a *natural experiment*: a dataset of individuals and effects, where we have knowledge of which individuals received treatment, but no control over how those treatments were assigned.

The natural experiment setting presents a number of challenges. Notably, treatments in a natural experiment can be assigned in a way that is correlated with the outcomes [CBGGB20, CCB+22]. So a direct comparison of the outcomes for the treatment and control groups would give a flawed estimate of the treatment effect. Perhaps even more difficult, the treatments can be correlated with the treatment effect itself [Lea19, dVKM+21]. Before giving a formal problem statement, we describe a real-world case study where these difficulties arise.

---

[1] `https://github.com/rtealwitter/naturalexperiments`

## 1.1 Reach Out and Read Colorado

Reach Out and Read Colorado[2] (RORCO) is a nonprofit organization that focuses on early childhood literacy in Colorado. They partner with health care clinics to provide books and a "prescription for reading to children" at regularly scheduled well-child visits. This model is based on a national Reach Out and Read[3] framework, the effectiveness of which has been evaluated in prior work, including via randomized control trials [Zuc09, ZK10, ZN20]. Prior studies confirm the positive effect of the general Reach Out and Read program on literacy outcomes, but primarily in urban areas in the northeastern United States. There have been no studies on the program in Colorado, and due to variations in implementation and differences in demographics, we should be cautious about extrapolating the findings of prior studies to RORCO [URH+23].

To address this gap, RORCO would like a stand-alone analysis of the effectiveness of its work. Of course, a longitudinal randomized control trial would give the most reliable results, but it would be time- and resource-intensive. Instead, we are interested in leveraging data that already exists from more than two decades of RORCO work. The goal is to combine this data on where and when RORCO provided books with publicly available information on students and schools in Colorado, including statistics on standarized tests that evaluate literacy.

Since not all students in the state of Colorado are treated in health clinics that partner with RORCO, this data provides a natural experiment, with only part of the population being exposed to the treatment we hope to analyze. One key challenges is that RORCO specifically directs treatments to under-served communities that simultaneously have the lowest literacy outcomes and which the program expects to experience the largest treatment effect.

Techniques for estimating treatment effect based on natural experiment data like the data available to RORCO have been widely studied for decades [RW00, Dun08, THM21, YHJ+22]. However, there is no clearly agreed upon best-practice method and, as we will show, estimators can return vastly different estimates of the impact of RORCO. We begin by formally describing the treatment effect problem.

## 1.2 Treatment Effect Estimation

Consider a set of $n$ observations. Each observation has $d$ covariates $\mathbf{x}_i \in \mathbb{R}^d$ where $i \in \{1, \ldots, n\} = [n]$. Every observation either receives the treatment or control, denoted by the treatment assignment $z_i \in \{0, 1\}$. Depending on whether the observation receives the treatment or control, we observe the treatment outcome $y_i^{(1)} \in \mathbb{R}$ or the control outcome $y_i^{(0)} \in \mathbb{R}$, but not both. The propensity score—the probability that an observation receives the treatment—is given by $p_i \in (0, 1)$. We make the standard assumption that treatments are assigned independently; i.e., that $p_i = \Pr(z_i = 1) = \Pr(z_i = 1 | z_j \forall j \neq i)$. In the problem, we are given the covariates, the treatment assignment, and the observed outcome but not the unobserved outcome or the propensity score. However, if the outcomes and propensity scores are related to the covariates as they often are, we can use the covariates to predict the unobserved outcome and the propensity score. The average treatment effect is defined as

$$\tau = \frac{1}{n} \sum_{i=1}^{n} \left( y_i^{(1)} - y_i^{(0)} \right).$$

The treatment effect problem is to create an estimator $\hat{\tau}$ that is close to $\tau$. The challenge is that, for each observation, the estimator can only use the treatment outcome $y_i^{(1)}$ or control outcome $y_i^{(0)}$, but not both. An estimator is unbiased if $\mathbb{E}[\hat{\tau}] = \tau$ where the expectation is with respect to the treatment assignment $\mathbf{z} \in \{0, 1\}^n$ and any internal randomness of the estimator. The goal is to build estimators that are unbiased and have small expected squared error $\mathbb{E}[(\hat{\tau} - \tau)^2]$. When an estimator is unbiased, its expected squared error is the variance $\mathrm{Var}(\hat{\tau} - \tau)$ so we will often refer to minimizing the variance of the estimator.

While we consider many estimators in this work, we focus on doubly robust estimators. Let $f^{(0)}, f^{(1)} : \mathbb{R}^d \to \mathbb{R}$ be learned functions for the control and treatment outcomes, respectively. Let $p_i$ be the

---

[2]https://reachoutandreadco.org/
[3]https://reachoutandread.org/

(estimated) propensity score for individual $i \in [n]$. Doubly robust estimators can be written as

$$\tau(\mathbf{z}) = \frac{1}{n} \sum_{i=1}^{n} \left( \frac{y_i^{(1)} - f^{(1)}(\mathbf{x}_i)}{p_i} 1[z_i = 1] - \frac{y_i^{(0)} - f^{(0)}(\mathbf{x}_i)}{1 - p_i} 1[z_i \neq 1] + f^{(1)}(\mathbf{x}_i) - f^{(0)}(\mathbf{x}_i) \right)$$

where different doubly robust estimators may be obtained by changing the way the functions $f^{(0)}$ and $f^{(1)}$ are learned, or including different estimates of the propensity scores.

Please refer to Appendix H for the other estimators we consider in our work.

## 1.3 Related Work

There are several popular datasets for treatment effect estimation. The Jobs dataset is a based on a small job training natural experiment where the control outcomes are incomes from before the training in 1975 and the treatment outcomes are incomes from after the training in 1978 [LaL86]. The Twins dataset is based on an observational study of twins that uses which twin is born heavier as the treatment, disregarding the difference in weight [ACL05]. The IHDP dataset is based on real covariates from an observational study starting in 1985 but with synthetic outcomes drawn from a normal distribution with a constant treatment effect (details in Section 4.1) [Hil11]. The News dataset is synthetically generated to broadly mimic a preference for reading on mobile devices, but without domain expert guidance (details in Section 6.2) [JSS16]. The ACIC dataset is synthetically generated for a competition, making strong assumptions on the outcomes (details in Section 2) [HDM19].

Because estimating treatment effect is an important task, there are many estimators that use a variety of techniques. We provide a brief description of prior work here and an expanded description in Appendix G. Some estimators use propensity scores to compare similar observations [Aus11, Lin14, AS15] and others use regression to adjust outcomes with predictions [Rho10, CKLP17, CT22]. Some estimators offer theoretical guarantees under certain assumptions [Fre08, BLB+09, Ken23] and others are robust to modelling errors [VdLR+11, VV15, Tan20]. Recently, there has been substantial work designing sophisticated neural network architectures and loss functions to estimate treatment effects [SBV19, KSBY19, CVdS21a]. Generally, however, the approaches are complicated and resource-intensive. In addition, as we will see on the RORCO data in Appendix D, the estimators can generate substantially different estimates on the same data.

In our experiments, we find that doubly robust estimators tend to outperform other methods. Asymptotically as the number of samples grows, doubly robust estimators are unbiased if the propensity scores are accurate *or* the outcome predictions are accurate [SRR99, KS07, VV15, Ken23]. We propose a new doubly robust algorithm called Double-Double which is most similar to the Off-policy estimator of Mou et al. [MWB22]. However, our method differs in two important ways: First, we separately learn the treatment and control outcomes. This is a more powerful variance reduction strategy that allows us to exactly analyze the variance of our estimator. Second, we learn the outcomes with a different loss function that stems from our more accurate variance analysis.

## 1.4 Our Contributions

Our contributions are four-fold.

1. We release the RORCO dataset, specifically designed for treatment effect estimation in an early childhood literacy natural experiment. The dataset includes observational outcomes (RORCO Real) and synthetic outcomes (RORCO) designed in consultation with literacy experts. We document and release the generation process and source data.

2. We create a comprehensive benchmark of more than 20 treatment effect estimators. The benchmark evaluates how the estimators perform as the sample size, propensity score accuracy, and correlation vary. In the benchmark experiments, we observe that doubly robust estimators often outperform the other methods, even by orders of magnitude.

3. We theoretically analyze doubly robust estimators, exactly deriving the finite-variance of any doubly robust estimator that uses splitting to obtain an unbiased estimate. Motivated by the analysis, we introduce Double-Double, a theoretically justified doubly robust estimator.

4. We release a Python package called `naturalexperiments` with our novel dataset, benchmark, and algorithm. The package also loads the Jobs, Twins, IHDP, News, and ACIC

| Dataset | Size | Variables | Treated % | BCE | $\text{Corr}(\mathbf{y}^{(1)}, \mathbf{p})$ | $\text{Corr}(\mathbf{y}^{(0)}, \mathbf{p})$ |
|---|---|---|---|---|---|---|
| JOBS | 722 | 8 | 41.1 | 0.0856 | 0.0355 | 0.0541 |
| TWINS | 50820 | 40 | 49.4 | 0.499 | -0.00311 | -0.0036 |
| IHDP | 747 | 26 | 18.6 | 0.452 | 0.0967 | 0.0236 |
| NEWS | 5000 | 3 | 45.8 | 0.545 | 0.86 | -0.565 |
| ACIC 2016 | 4802 | 54 | 18.4 | 0.372 | 0.112 | 0.0383 |
| ACIC 2017 | 4302 | 50 | 47.4 | 0.436 | -0.269 | -0.153 |
| RORCO Real | 4178 | 78 | 25.3 | 0.158 | -0.000602 | -0.0739 |
| RORCO | 21663 | 78 | 44.3 | 0.212 | -0.986 | -0.989 |

Table 1: Comparison of dataset attributes. Size is the number of observations, Variables is the number of variables, Treated is the percent of observations that receive the treatment, BCE is binary cross entropy between the propensity scores and treatment assignment, $\text{Corr}(\mathbf{y}^{(1)}, \mathbf{p})$ is Pearson's correlation coefficient between the treatment outcomes and propensity scores, and $\text{Corr}(\mathbf{y}^{(0)}, \mathbf{p})$ is Pearson's correlation coefficient between the control outcomes and propensity scores.

datasets and is built to facilitate the easy addition of new estimators and datasets. Appendix C shows the code used to generate (almost all of) the results in the paper and appendices.

## 2 RORCO Dataset

The RORCO dataset describes student literacy and participation in the RORCO nonprofit program. Due to privacy concerns, the observations in the dataset are grades. For example, one observation corresponds to the third grade class at Academy Endeavor Elementary School in the 2018-19 school year. The covariates include variables like student counts, student-to-teacher ratios, demographic information, instructional programs, socioeconomic status indicators like free and reduced lunch eligibility, attendance rates, and staff information like average salary. A summary of the covariates appears in Appendix K and detailed documentation of the process to create the dataset is available on the Github repository.[4] Using the covariates, we created two different versions of the dataset: an observational version we call RORCO Real and a semi-synthetic version we call RORCO.

### 2.1 RORCO Real: An Observational Dataset

The observational RORCO Real dataset uses real literacy outcomes: The Colorado Measures of Academic Success, known as CMAS, is the state's common measurement of students' progress at the end of the school year. Because of the effect of COVID-19 on well-child visits and education [BBME22, KF22], we restricted our data to literacy outcomes between 2014 and 2019. (CMAS was only fully implemented in Spring 2014.)

We determined which observations received the treatment via a RORCO dataset. The dataset included the number of well-child visits where books were given out by age for each clinic in Colorado and in each year. At the suggestion of RORCO, we made the assumption that a child *in a rural area* who received a book from a RORCO well-child visit attended the nearest school when they reached school age. We then marked an observation as receiving RORCO treatment if more than half the students in the class received a RORCO well-child visit under this assumption. Because of the proximity assumption, we restricted the RORCO Real dataset to only rural clinics and schools.

Figure 1 shows the treatment and control outcomes by propensity score in the RORCO Real version. Because of the strong assumption in the data generation process, we expect substantial noise. Nonetheless it is clear that RORCO has a positive treatment effect for the majority of observations, especially as the likelihood of receiving the treatment increases.

Table 4 in Appendix D shows the estimate on the real outcomes and treatments for each estimator in the benchmark. The estimators return surprisingly different results. In order to evaluate which estimators are accurate, we create a semi-synthetic dataset with treatment and control outcomes.

---

[4]`https://github.com/rtealwitter/naturalexperiments/blob/main/naturalexperiments/data/rorco/rorco_documentation.md`

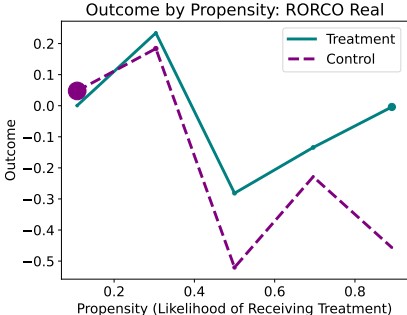

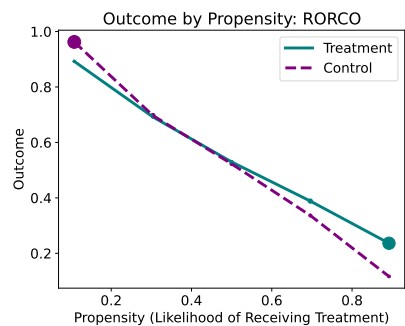

Figure 1: Normalized CMAS scores for five equal sized groups plotted against their propensity scores. The treatment has more effect on children who are likely to receive the treatment.

Figure 2: Synthetic outcomes designed in consultation with domain experts: Treatments are targeted to under-served children who benefit more [Neu99, NC01].

## 2.2 RORCO: A Semi-synthetic Dataset

For the observational RORCO dataset, we developed synthetic outcomes and treatments in consultation with early childhood literacy experts. The literacy experts suggested the following assumptions:

(1) The outcomes should be inversely related to the propensity score. That is, students are more likely to participate in the literacy program if they have lower literacy proficiency [Neu99].

(2) The treatment effect should be negligible for observations with small propensity scores and increasing for larger propensities. That is, students who are less likely to participate in the program, and so have higher literacy proficiency as per (1), will not benefit from the program because they already have sufficient resources. In contrast, students who are more likely to participate in the program, and so have lower literacy proficiency as per (1), will benefit from the program in proportion to their literacy needs [NC01].

Based on (1) and (2), we made the control outcomes as depicted in Figure 2 vary between 0 and 1 with an inverse linear relationship to propensity scores. We made the treatment outcomes align with the control outcomes until .5 and then increasingly separate (with a negated square root added to the control outcomes).

The RORCO outcomes reflect the suggestions of literacy experts while the RORCO Real outcomes are based on a best guess based proximity connection between well-child visits and standardized test scores. We intentionally make the synthetic outcomes reflect the guidance of the literacy experts rather than tailoring to the real (noisy) outcomes we observe. We believe the different outcomes are a benefit, making our benchmark more robust.

## 3 Benchmark

We evaluate more than 20 treatment effect estimators on the RORCO dataset. Since computing the true treatment effect requires both treatment and control outcomes, we use the RORCO dataset with synthetic outcomes. Since they are almost never known in practice, we estimate the propensity scores from the data. In addition, we regularize the estimated propensity scores by truncating them to the range $[.01, .99]$. As shown by Figure 3, the propensity scores are well-calibrated. In 100 runs on RORCO, we find the cross entropy between the true propensities and a sampled treatment assignment is $.202 \pm .029$ while the cross entropy between the predicted propensities and a sampled treatment assignment vector is $.196 \pm .029$. This suggests that the predicted propensity scores are quite accurate, at least on the synthetic data where we know the true propensity scores.

Due to space constraints, the estimators are described in detail in Appendix H. For the estimators that are agnostic to the learning process, we use a three-layer neural network with 100 hidden nodes and ReLU activations after all intermediate layers, .001 learning rate, and 200 epochs. We include experiments with other learning models (e.g., BART and causal forests) in Appendix L. In contrast, all of the "Net" estimators have custom neural network architectures and we use the CATENet

benchmark [5] [CVdS21a, CVdS21b] implementation. All experiments are run on a cluster of 24-core Intel Cascade Lake Platinum 8268 chips. Table 2 displays the squared error on the semi-synthetic RORCO dataset over 100 runs. Due to space constraints, we include the analogous tables for the ACIC 2016, ACIC 2017, IHDP, Jobs, News, and Twins datasets in Appendix J. Because some of the CATENet estimators are slow to compute, we subsample to 5000 observations in our experiments unless otherwise stated.

With the exception of the Twins dataset, the standard doubly robust estimator produces the lowest empirical mean squared error followed by Double-Double. After Double-Double, several CATENet estimators—FlexTENet, TNet, TARNet, and RANet—give the best performance; however, they require substantially more training time. Next, we examine how the estimators perform as the number of observations, accuracy of the propensity scores, and correlation between treatment and outcomes varies. The goal is to evaluate the estimators in different realistic settings.

### 3.1 Squared Error by Number of Observations

The number of observations is a fixed component of real experiments. In some settings, there may be fewer observations because administering the treatment or collecting data is resource intensive or infeasible. We investigate how the estimators perform as the number of observations varies.

The plots show the squared error between the true treatment effect and estimated treatment effect on a logarithmic scale. We run each experiment 100 times; the lines indicate the median and the shaded intervals indicate the region within the first and third quartiles. So that they remain legible, we restrict the plots to the six best performing estimators in Table 2. Figure 4 compares the estimators as a function of the number of observations. Since each estimator has a learning component, performance improves with the number of observations.

---

[5] github.com/AliciaCurth/CATENets

| Method | Mean | 1st Quartile | 2nd Quartile | 3rd Quartile | Time (s) |
|---|---|---|---|---|---|
| Regression Discontinuity | 4.65e-03 | 2.72e-03 | 3.84e-03 | 5.52e-03 | 9.55e-04 |
| Propensity Stratification | 2.57e-03 | 1.52e-03 | 2.25e-03 | 3.29e-03 | 2.78e-03 |
| Direct Difference | 4.48e-01 | 3.57e-01 | 4.18e-01 | 5.79e-01 | 4.74e-04 |
| Adjusted Direct | 6.29e-03 | 5.25e-03 | 6.20e-03 | 7.14e-03 | 1.15e+01 |
| Horvitz-Thompson | 1.06e-02 | 4.29e-03 | 9.20e-03 | 1.44e-02 | 4.65e-04 |
| TMLE | 1.19e-01 | 7.21e-03 | 2.60e-02 | 7.43e-02 | 2.35e+01 |
| Off-policy | 3.17e-03 | 1.86e-03 | 2.86e-03 | 4.11e-03 | 1.14e+01 |
| Double-Double | 1.07e-05 | 1.06e-06 | 4.41e-06 | 1.45e-05 | 2.29e+01 |
| Doubly Robust | 9.98e-07 | 1.48e-07 | 5.42e-07 | 1.37e-06 | 9.89e+00 |
| Direct Prediction | 1.36e-02 | 3.60e-03 | 1.02e-02 | 1.94e-02 | 1.23e+01 |
| SNet | 2.57e-02 | 4.85e-03 | 1.21e-02 | 3.62e-02 | 3.49e+01 |
| FlexTENet | 1.15e-03 | 4.28e-05 | 1.09e-04 | 4.95e-04 | 1.56e+02 |
| OffsetNet | 1.10e-03 | 7.72e-04 | 9.90e-04 | 1.41e-03 | 1.30e+02 |
| TNet | 8.05e-04 | 6.39e-05 | 2.50e-04 | 4.37e-04 | 1.06e+02 |
| TARNet | 1.92e-04 | 2.70e-05 | 1.04e-04 | 2.38e-04 | 1.01e+02 |
| DragonNet | 2.18e-02 | 4.42e-03 | 1.71e-02 | 2.46e-02 | 6.88e+00 |
| SNet3 | 1.80e-02 | 3.48e-03 | 9.80e-03 | 2.50e-02 | 2.36e+01 |
| DRNet | 5.00e-03 | 1.53e-04 | 6.01e-04 | 2.25e-03 | 1.14e+02 |
| RANet | 7.85e-04 | 3.67e-05 | 2.08e-04 | 7.06e-04 | 1.91e+02 |
| PWNet | 2.28e-01 | 7.02e-03 | 4.00e-02 | 2.82e-01 | 1.13e+02 |
| RNet | 2.96e-03 | 2.47e-03 | 2.84e-03 | 3.43e-03 | 5.83e+01 |
| XNet | 1.00e-03 | 3.08e-05 | 2.29e-04 | 9.26e-04 | 2.41e+02 |

Table 2: Squared error on the semi-synthetic RORCO dataset. The summary statistics are computed over 100 runs. The randomness in the runs comes from the synthetically generated outcomes, estimates of the propensity scores, and any internal randomness in the estimators. Note that we adopt the Olympic medal convention: gold , silver and bronze cell highlights signify first, second and third best performance, respectively.

## 3.2 Squared Error by Correlation

Correlation between outcomes and propensity scores is a challenging component of natural experiments. We investigate how the correlation affects the performance of the estimators. We measure correlation using distance correlation [SRB07]. Unlike the Pearson correlation coefficient which is mainly sensitive to a linear relationship [Pea95], the distance correlation is zero if and only if the random variables are independent. We opt for the distance correlation instead of Spearman's rank correlation because the propensity scores are concentrated close to 0 and 1, making the rank brittle to small perturbations [Spe87].

In Figure 5, we add noise to the outcomes and compute the distance correlation. The plot shows the squared error against the average of the distance correlation between propensity scores $\mathbf{p}$ and treatment outcomes $\mathbf{y^{(1)}}$ and the distance correlation between $\mathbf{p}$ and control outcomes $\mathbf{y^{(0)}}$. For all estimators, the squared error generally decreases as the distance correlation increases. The doubly robust estimator and Double-Double outperform the other estimators until the distance correlation surpasses .8.

## 3.3 Squared Error by Propensity Accuracy

Since propensity scores are almost never known, estimating propensity scores is an important part of treatment effect estimation. We investigate how the accuracy of the propensity scores affects the performance of the estimators. We add noise to the propensity scores and then compute the cross entropy between the noised propensity scores and the observations that receive treatment as a measure of inaccuracy. Since the CATENet estimators do not rely on externally computed propensity scores, we consider the six best non-CATENet estimators.

Figure 6 shows the squared error against propensity score accuracy as measured by cross entropy. Because of its propensity score weighting in the loss function, Double-Double is quite sensitive to propensity accuracy. While it performs the best when the propensity scores are accurate, the doubly robust estimator still remains competitive as the propensity scores degrade.

## 4 Doubly Robust Analysis

Because of their superior performance in the benchmark, we theoretically analyze a broad-class of doubly robust estimators. The standard doubly robust estimator uses each observation to both learn and evaluate the same predictive function, introducing complicated statistical dependence. Instead, we consider doubly robust estimators with split training, ensuring that the prediction for each observation is independent of its outcomes [MWB22]. We show that such estimators are unbiased and we exactly derive their finite-variance. Such estimators have been analyzed in prior work but, to

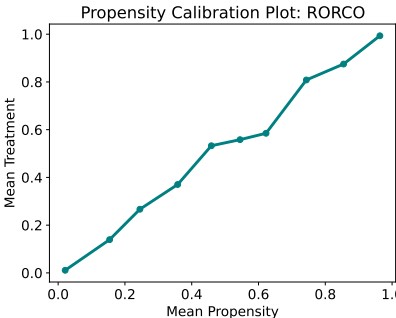

Figure 3: Mean treatment rate and mean propensity score among observations with similar propensity scores. Because the predicted and actual treatment rates are close to the identity line, we conclude the propensity scores are well calibrated.

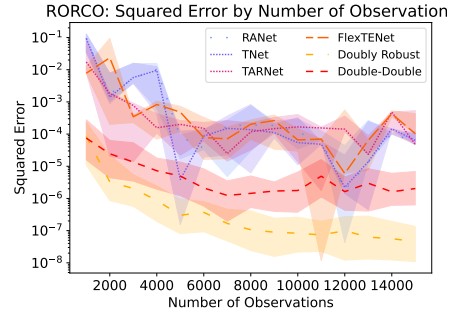

Figure 4: Squared error of each estimator by the number of observations. The darker line is the median and the shaded region encompasses the first and third quartile across 100 runs. The doubly robust estimator, followed by Double-Double, achieve the lowest squared error.

**Algorithm 1** Doubly Robust Estimator with Split Training

---

**Input:** Covariates $\mathbf{X} \in \mathbb{R}^{n \times d}$, treatment $\mathbf{z} \in \{0,1\}^n$, outcomes $\mathbf{y}^{(1)} \in \mathbb{R}^n$ and $\mathbf{y}^{(0)} \in \mathbb{R}^n$, propensity scores $\mathbf{p} \in (0,1)^n$, a treatment weights $\mathbf{w}^{(1)} \in \mathbb{R}^n$, and control weights $\mathbf{w}^{(0)} \in \mathbb{R}^n$
Randomly partition the data into sets $S_1$ and $S_2$
**for** $j \in \{1, 2\}$ **do**
    Learn $\hat{f}_{\mathbf{z},j}^{(1)}$ to minimize the treatment loss $\sum_{i \in S_j : z_j = 1}(y_i^{(1)} - f(\mathbf{x}_i))^2 w_i^{(1)}$
    Learn $\hat{f}_{\mathbf{z},j}^{(0)}$ to minimize the control loss $\sum_{i \in S_j : z_j \neq 1}(y_i^{(0)} - f(\mathbf{x}_i))^2 w_i^{(0)}$
    **for** $i \notin S_j$ **do**
        Compute the adjustment $\hat{y}_i(\mathbf{z}) = (1 - p_i)\hat{f}_{\mathbf{z},j}^{(1)}(\mathbf{x}_i) + p_i \hat{f}_{\mathbf{z},j}^{(0)}(\mathbf{x}_i)$
    **end for**
**end for**
**return** the estimator $\hat{\tau}(\mathbf{z})$ where

$$\hat{\tau}(\mathbf{z}) = \frac{1}{n}\sum_{i=1}^{n}\left(\frac{y_i^{(1)} - \hat{y}_i(\mathbf{z})}{p_i}\mathbb{1}_{z_i=1} - \frac{y_i^{(0)} - \hat{y}_i(\mathbf{z})}{1 - p_i}\mathbb{1}_{z_i \neq 1}\right) \tag{2}$$

---

the best of our knowledge, all prior results are upper bounds as opposed to exact characterizations of the finite-variance. We then introduce Double-Double, a doubly robust estimator motivated by the exact variance expression.

Recall the standard doubly robust estimator is given by

$$\hat{\tau}(\mathbf{z}) = \frac{1}{n}\sum_{i=1}^{n}\left(\frac{y_i^{(1)} - f(\mathbf{x}_i)^{(1)}}{p_i}\mathbb{1}_{z_i=1} - \frac{y_i^{(0)} - f(\mathbf{x}_i)^{(0)}}{1 - p_i}\mathbb{1}_{z_i \neq 1} + f(\mathbf{x}_i)^{(1)} - f(\mathbf{x}_i)^{(0)}\right) \tag{1}$$

where $f^{(1)}, f^{(0)} : \mathbb{R}^d \to \mathbb{R}$ are learned function of the covariates. The estimator will have complicated statistical dependencies if the learned function is applied to the observations in the training set. Instead, we consider doubly robust estimators with split training as described in Algorithm 1. The formulation of the estimator in the pseudocode is different from the standard notation, making it easier to present the variance results. In Appendix F, we show that the expressions in Equations 1 and 2 are equivalent for appropriately defined learned functions.

In the standard doubly robust estimator, the training weights in Algorithm 1 are all 1 i.e., $w_i^{(1)} = w_i^{(0)} = 1$. We will explore how to choose the weights so as to minimize the finite-variance as derived in Theorem 4.1. For the notation in the theorem statement, let $\mathbf{z}^{(j \to b)}$ be the assignment vector with $z_j$ set to b.

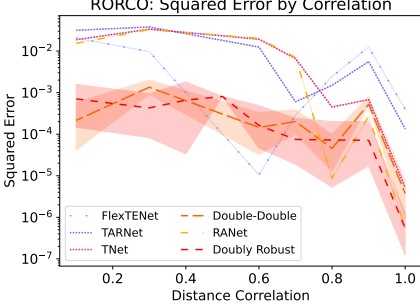

Figure 5: Squared error by distance correlation. The doubly robust estimator and Double-Double outperform the other estimators until the distance correlation surpasses .8.

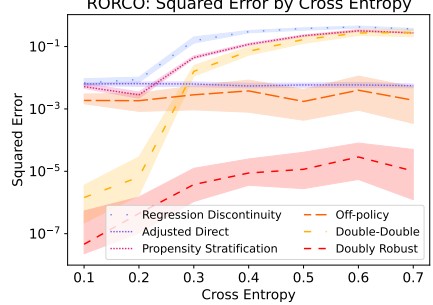

Figure 6: Squared error by the cross entropy between the estimated propensity scores and the treatment assignment. Double-Double is sensitive to propensity score accuracy whereas the doubly robust estimator is, well, robust.

| Method | Mean | 1st Quartile | 2nd Quartile | 3rd Quartile | Time (s) |
|---|---|---|---|---|---|
| Doubly Robust | 9.98e-07 | 1.48e-07 | 5.42e-07 | 1.37e-06 | 9.89e+00 |
| DR + Weighting | 4.02e-06 | 5.46e-07 | 2.62e-06 | 5.57e-06 | 9.81e+00 |
| DR + 2x Weighting | 3.80e-06 | 2.48e-07 | 9.71e-07 | 3.82e-06 | 9.80e+00 |
| DR + Split | 9.82e-05 | 3.27e-06 | 1.21e-05 | 3.65e-05 | 2.22e+01 |
| DR + Split + Weight | 1.12e-04 | 2.19e-06 | 1.03e-05 | 2.41e-05 | 2.22e+01 |
| Double-Double | 1.07e-05 | 1.06e-06 | 4.41e-06 | 1.45e-05 | 2.29e+01 |

Table 3: Ablation results for doubly robust estimators on the RORCO dataset. The doubly robust estimators without the training split give the best performance, likely because they effectively have access to twice the data in the training process. However, for doubly robust estimators with the training split, the theoretically justified Double-Double gives the best performance.

**Theorem 4.1.** *When the propensity scores are known exactly, the doubly robust estimator with split training $\hat{\tau}(\mathbf{z})$ is unbiased i.e., $\mathbb{E}_{\mathbf{z},S_1,S_2}[\hat{\tau}(\mathbf{z}) - \tau] = 0$ with variance given by*

$$\text{Var}[\hat{\tau}(\mathbf{z}) - \tau] = \frac{1}{n^2}\sum_{i=1}^{n}\mathbb{E}_{\mathbf{z},S_1,S_2}\left[\left((y_i^{(1)} - \hat{f}_{\mathbf{z},S(i)}^{(1)}(\mathbf{x}_i))\sqrt{\frac{1-p_i}{p_i}} + (y_i^{(0)} - \hat{f}_{\mathbf{z},S(i)}^{(0)}(\mathbf{x}_i))\sqrt{\frac{p_i}{1-p_i}}\right)^2\right]$$

$$+ \frac{1}{n^2}\sum_{i\neq j}\mathbb{E}_{\mathbf{z},S_1,S_2}\left[\left(\hat{y}_i(\mathbf{z}^{(j\to 1)}) - \hat{y}_i(\mathbf{z}^{(j\to 0)})\right)\left(\hat{y}_j(\mathbf{z}^{(i\to 1)}) - \hat{y}_j(\mathbf{z}^{(i\to 0)})\right)\right].$$

The proof of Theorem 4.1 appears in Appendix E. We now discuss how to choose the weights and train the learned functions so as to minimize the variance terms.

The first variance term captures the weighted difference between the outcomes and predictions. Unfortunately, minimizing the loss function directly is not possible because only $y_i^{(1)}$ or $y_i^{(0)}$ is known for any given observation $i$. Instead, we can minimize an upper bound on the variance.

**Weighting** If we choose weights $w_i^{(1)} = \frac{1-p_i}{p_i}$ and $w_i^{(0)} = \frac{p_i}{1-p_i}$, then the loss functions reflect the first variance term. Intuitively, the weights prioritize correct predictions on observations that are less likely to be seen, ensuring that the learned function is accurate for all propensity scores. If $p_i$ is small but $z_i = 1$, then $w_i^{(1)}$ is quite large. However, there is an additional bias which is not yet accounted for: whether an observation appears in the training set depends on its propensity.

**Double Weighting** If we choose weights $w_i^{(1)} = \frac{1-p_i}{p_i^2}$ and $w_i^{(0)} = \frac{p_i}{(1-p_i)^2}$, then the *expected* loss functions reflect the first variance term. In particular, the expectation of the treatment loss in set $S_j$ is

$$\mathbb{E}_{\mathbf{z}}\left[\sum_{i\notin S_j}\mathbb{1}_{z_i=1}\frac{1-p_i}{p_i^2}(y_i^{(1)} - f_{\mathbf{z},j}^{(1)}(\mathbf{x}_i))^2\right] = \sum_{i\notin S_j}\frac{1-p_i}{p_i}\mathbb{E}_{\mathbf{z}}[(y_i^{(1)} - f_{\mathbf{z},j}^{(1)}(\mathbf{x}_i))^2].$$

Over both loss functions and both sets $S_1$ and $S_2$, the total expected loss upper bounds the first variance term by the AM-GM inequality: for any real numbers $a$ and $b$, $(a+b)^2 = a^2 + 2ab + b^2 \leq 2a^2 + 2b^2$

Motivated by the upper bound on the variance term, we introduce Double-Double: a doubly robust estimator with double weighting. Double-Double is equivalent to Algorithm 1 with $w_i^{(1)} = \frac{1-p_i}{p_i^2}$ and $w_i^{(0)} = \frac{p_i}{(1-p_i)^2}$. In Table 3, we observe that Double-Double gives the best performance of the doubly robust estimators with the training split. However, perhaps because of the additional training data available, doubly robust estimators without the training split perform better.

The second variance term measures function sensitivity to removing or adding observations to the training set, a quantity closely related to differential privacy (DP). In Appendix I, we explore DP learning but find no improvement on the mean squared error. We believe the reason is that the second term is quite small in practice: On the RORCO dataset, we find that the second term is roughly $10^{-30}$.

When the propensity scores are independent of the outcomes and covariates, the expectation of the weighted loss function is proportional to the expectation of the unweighted loss function. Suppose

the data is drawn from a distribution $\mathcal{D}$. Then, if the propensities are independent of the outcomes,

$$\mathbb{E}\left[\frac{1}{n}\sum_{i=1}^{n}\frac{1-p_i}{p_i}(y_i^{(1)}-f_{\mathbf{z},j}^{(1)}(\mathbf{x}_i))^2\right] = \mathbb{E}_{\mathcal{D}}\left[\frac{1-p}{p}(y^{(1)}-f_{\mathbf{z},j}^{(1)}(\mathbf{x}))^2\right]$$

$$= \mathbb{E}_{\mathcal{D}}\left[\frac{1-p}{p}\right]\mathbb{E}_{\mathcal{D}}\left[(y^{(1)}-f_{\mathbf{z},j}^{(1)}(\mathbf{x}))^2\right].$$

The analogous equalities follow for the control loss. So the weighted loss functions reduce to unweighted loss functions in the standard setting where the propensity scores are independent of the outcomes and covariates.

## 5    Limitations and Conclusion

We made several assumptions while building the RORCO and RORCO Real datasets. For the RORCO Real dataset, we made assumptions in order to determine whether a class (the most granular education data available) received the RORCO "treatment". These assumptions potentially bias the resulting datasets in the following ways: By using class (as opposed to individual students) as observations, we potentially reduce our ability to measure the effect RORCO. For example, if only half the class received the RORCO treatment then the effect on their literacy outcomes will be weaker. By using proximity in rural communities to determine whether classes received the RORCO treatment, we change the distribution of the data to only reflect sparsely populated geographic areas. Further, we make an unverified assumption that students did not move over the course of several years in these rural communities, i.e., they attend school near where they lived as a child. For the RORCO dataset, we made assumptions to synthetically generate outcomes. While conforming to expert understanding, these assumptions do not necessarily reflect what happens in the real world. As a result, fine-tuning estimators only on these synthetic outcomes may result in algorithms that are not applicable to real settings.

In Section 4, we analyzed doubly robust estimators with a testing-training split and proposed a theoretically motivated estimator called Double-Double. While perhaps slightly unsatisfying that the non-splitting methods perform better than Double-Dobule, this is not surprising, as they effectively have access to twice the data. A natural question for future work would be a full analysis of the non-splitting method. In the analysis, we assumed that the propensity scores are known exactly. Understanding how robust the variance analysis is to propensity score accuracy is an important direction for future work.

We introduce RORCO, a novel and reproducible dataset showcasing the unique challenges of treatment effect estimation in natural experiments. We release RORCO and an extensive benchmark of more than 20 treatment effect estimators in the `naturalexperiments` package. From the benchmark on RORCO and six additional datasets, we find that doubly robust estimators often perform the best in natural experiments. Our theoretical analysis sheds light on their performance and motivates the Double-Double estimator. Our work is not without limitations, the observational version of RORCO makes a strong assumption on clinic-school proximity to determine treatment since we cannot track individuals and our theoretical analysis applies only to doubly robust estimators with a training split. While algorithms can be misused, we believe our work will result in a net positive impact because of its highly specialized nature.

## Acknowledgements

This work was supported by the National Science Foundation under Grant No. 2045590 and Graduate Research Fellowship Grant No. DGE-2234660.

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

## B  Dataset Supplement

For a description of the RORCO dataset, please refer to Section 2. The dataset is intended broadly for academic research, in particular designing estimators for causal inference. We hope the RORCO dataset encourages more accurate estimators and more comprehensive evaluations. The dataset is available in the `naturalexperiments` Python package.[6] The dataset can be directly downloaded as a CSV from the Github repository.[7] Alternatively, after installing the `naturalexperiments` package with pip, the covariates, outcomes, and treatment assignments can be directly loaded in Python with the following code.

```python
import naturalexperiments as ne

# Semi-synthetic version
X, y, z = ne.dataloaders["RORCO"]()

# Observational version
X, y, z = ne.dataloaders["RORCO Real"]()
```

Repeated calls to the observational dataloader return the same outcomes and treatment assignment whereas repeated calls to the semi-synthetic dataloader return newly generated outcomes and treatment assignments. We provide an introductory demonstration with examples of the many tools in `naturalexperiments` in the Github repository and, in Appendix C, we showcase almost all of the code used to produce the results in the paper and appendices. The Croissant metadata is available on the Github repository.[8] The authors bear all responsibility in case of a violation of rights. We confirm the use of the MIT license.[9] The code and data will be stored and maintained indefinitely on the Github repository. Interested researchers may email us directly or open issues. When the Dataset nutrition label is approved, we will update the README.md file on the Github repository.

---

[6]`https://github.com/rtealwitter/naturalexperiments`

[7]`https://raw.githubusercontent.com/rtealwitter/naturalexperiments/main/naturalexperiments/data/rorco/rorco_data.csv`

[8]`https://github.com/rtealwitter/naturalexperiments/blob/main/naturalexperiments/data/rorco/metadata.json`

[9]`https://github.com/rtealwitter/naturalexperiments/blob/main/LICENSE`

## C  Using `naturalexperiments`

The following is the code used to produce almost all of the tables and figures in the paper and appendices. We exclude the code to produce the ablation and the differential privacy heatmaps because it is slightly longer.

```python
import naturalexperiments as ne

# Dataset summary
ne.dataset_table(ne.dataloaders)

# Dataset plots e.g., outcome by propensity, calibration, etc.
ne.plot_all_data(ne.dataloaders)

# Estimates on RORCO Real
variance, times = ne.compute_estimates(methods, "RORCO Real", num_runs
                                    =100)
ne.benchmark_table(variance, times)

# Benchmark
for dataset in ["ACIC 2016", "ACIC 2017", "IHDP", "JOBS", "NEWS", "
                                    TWINS", "RORCO"]:
    variance, times = ne.compute_variance(ne.methods, dataset,
                                        num_runs=100)
    ne.benchmark_table(variance, times)

# Benchmark by number of observations
use_methods = ["FlexTENet", "TNet", "TARNet", "RANet", "Double-Double"
                                    , "Doubly Robust"]
methods = {method: ne.methods[method] for method in use_methods}
ns = list([x * 1000 for x in range(1, 16)])
variance = ne.compute_variance_by_n(methods, "RORCO", ns=ns, num_runs=
                                    100)
ne.plot_estimates(variance, xlabel = "Number of Observations")

# Benchmark by correlation
variance = ne.compute_variance_by_correlation(methods, "RORCO",
                                    num_runs=100)
ne.plot_estimates(variance, xlabel = "Distance Correlation")

# Benchmark by cross entropy
use_methods = ["Regression Discontinuity", "Propensity Stratification"
                                    , "Adjusted Direct", "Off-policy",
                                    "Double-Double", "Doubly Robust"]
methods = {method: ne.methods[method] for method in use_methods}
variance = ne.compute_variance_by_entropy(methods, "RORCO", num_runs=
                                    100)
ne.plot_estimates(variance, xlabel = "Cross Entropy")
```

## D   RORCO Real Estimates

| Method | Mean | 1st Quartile | 2nd Quartile | 3rd Quartile | Time (s) |
|---|---|---|---|---|---|
| Regression Discontinuity | 1.54e-01 | 1.12e-01 | 1.65e-01 | 2.08e-01 | 8.61e-04 |
| Propensity Stratification | -2.90e-01 | -4.24e-01 | -2.93e-01 | -1.71e-01 | 2.63e-03 |
| Direct Difference | -8.78e-02 | -8.78e-02 | -8.78e-02 | -8.78e-02 | 4.60e-04 |
| Adjusted Direct | 1.55e-02 | -2.04e-02 | 1.06e-02 | 4.33e-02 | 1.10e+01 |
| Horvitz-Thompson | -5.84e-02 | -8.34e-02 | -6.36e-02 | -3.15e-02 | 4.40e-04 |
| Doubly Robust | -3.91e-02 | -6.62e-02 | -4.19e-02 | -1.60e-02 | 1.49e+01 |
| TMLE | 2.34e-01 | -1.28e+00 | 4.46e-02 | 1.77e+00 | 2.18e+01 |
| Off-policy | -1.38e-02 | -3.49e-02 | -1.33e-02 | 3.79e-03 | 1.12e+01 |
| Double-Double | -5.72e-02 | -8.19e-02 | -6.28e-02 | -2.78e-02 | 2.22e+01 |
| Direct Prediction | -2.23e-02 | -6.71e-02 | -2.53e-02 | 2.49e-02 | 1.12e+01 |
| SNet | -5.93e-02 | -5.93e-02 | -5.93e-02 | -5.93e-02 | 4.04e+01 |
| FlexTENet | 2.99e-02 | 2.99e-02 | 2.99e-02 | 2.99e-02 | 2.84e+01 |
| OffsetNet | 3.34e-02 | 3.34e-02 | 3.34e-02 | 3.34e-02 | 7.42e+00 |
| TNet | 5.43e-02 | 5.43e-02 | 5.43e-02 | 5.43e-02 | 7.84e+00 |
| TARNet | -2.90e-02 | -2.90e-02 | -2.90e-02 | -2.90e-02 | 6.79e+00 |
| DragonNet | 5.36e-03 | 5.36e-03 | 5.36e-03 | 5.36e-03 | 9.33e+00 |
| SNet3 | -8.72e-03 | -8.72e-03 | -8.72e-03 | -8.72e-03 | 3.43e+01 |
| DRNet | -2.05e-02 | -2.05e-02 | -2.05e-02 | -2.05e-02 | 1.45e+01 |
| RANet | -7.56e-03 | -7.56e-03 | -7.56e-03 | -7.56e-03 | 1.23e+01 |
| PWNet | -1.29e-01 | -1.29e-01 | -1.29e-01 | -1.29e-01 | 1.45e+01 |
| RNet | 3.62e-02 | 3.62e-02 | 3.62e-02 | 3.62e-02 | 1.05e+01 |
| XNet | 9.33e-02 | 9.33e-02 | 9.33e-02 | 9.33e-02 | 1.90e+01 |

Table 4: Estimates on the RORCO Real dataset. The outcomes are normalized: mean-centered and divided by the standard deviation. There is surprising variation in the estimates from the lowest mean estimate of $-.607$ to the highest of $.459$. The randomness in the 100 runs comes from the following sources: The propensity scores are generated from a learning process that uses random batches for training, the learned predictions (in most of the estimators) are also generated from a learning process that uses batches, and, in some estimators like Double-Double, there is a random training-testing split.

# E   Double-Double Variance

**Theorem 4.1.** *When the propensity scores are known exactly, the doubly robust estimator with split training $\hat{\tau}(\mathbf{z})$ is unbiased i.e., $\mathbb{E}_{\mathbf{z},S_1,S_2}[\hat{\tau}(\mathbf{z}) - \tau] = 0$ with variance given by*

$$\mathrm{Var}[\hat{\tau}(\mathbf{z}) - \tau] = \frac{1}{n^2} \sum_{i=1}^{n} \mathbb{E}_{\mathbf{z},S_1,S_2}\left[ \left( (y_i^{(1)} - \hat{f}_{\mathbf{z},S(i)}^{(1)}(\mathbf{x}_i)) \sqrt{\frac{1-p_i}{p_i}} + (y_i^{(0)} - \hat{f}_{\mathbf{z},S(i)}^{(0)}(\mathbf{x}_i)) \sqrt{\frac{p_i}{1-p_i}} \right)^2 \right]$$

$$+ \frac{1}{n^2} \sum_{i \neq j} \mathbb{E}_{\mathbf{z},S_1,S_2}\left[ \left( \hat{y}_i(\mathbf{z}^{(j \to 1)}) - \hat{y}_i(\mathbf{z}^{(j \to 0)}) \right)\left( \hat{y}_j(\mathbf{z}^{(i \to 1)}) - \hat{y}_j(\mathbf{z}^{(i \to 0)}) \right) \right].$$

*Proof.* To simplify notation in the proof, we will drop the subscript on the expectation and variance. Recall the estimator is given by

$$\hat{\tau}(\mathbf{z}) = \frac{1}{n} \sum_{i=1}^{n} \left( \frac{y_i^{(1)} - \hat{y}_i(\mathbf{z})}{p_i} \mathbb{1}_{z_i=1} - \frac{y_i^{(0)} - \hat{y}_i(\mathbf{z})}{1-p_i} \mathbb{1}_{z_i \neq 1} \right).$$

By linearity of expectation we have:

$$\mathbb{E}[\hat{\tau}(\mathbf{z})] = \frac{1}{n} \sum_{i=1}^{n} \mathbb{E}\left[ \frac{y_i^{(1)} - \hat{y}_i(\mathbf{z})}{p_i} \mathbb{1}_{z_i=1} \right] - \mathbb{E}\left[ \frac{y_i^{(0)} - \hat{y}_i(\mathbf{z})}{1-p_i} \mathbb{1}_{z_i \neq 1} \right].$$

Then, since the prediction $\hat{y}_i(\mathbf{z})$ is independent of the treatment assignment $z_i$, we can use the fact that $\mathbb{E}[AB] = \mathbb{E}[A]\,\mathbb{E}[B]$ for indendent random variables $A, B$ to obtain:

$$\mathbb{E}[\hat{\tau}(\mathbf{z})] = \frac{1}{n} \sum_{i=1}^{n} \mathbb{E}[y_i^{(1)} - \hat{y}_i(\mathbf{z})]\,\mathbb{E}[\mathbb{1}_{z_i=1}]/p_i - \mathbb{E}[y_i^{(0)} - \hat{y}_i(\mathbf{z})]\,\mathbb{E}\,\mathbb{1}_{z_i \neq 1}/(1-p_i)$$

$$= \frac{1}{n} \sum_{i=1}^{n} y_i^{(1)} - \mathbb{E}[\hat{y}_i(\mathbf{z})] - (y_i^{(0)} - \mathbb{E}[\hat{y}_i(\mathbf{z})]$$

$$= \frac{1}{n} \sum_{i=1}^{n} y_i^{(1)} - y_i^{(0)} = \tau.$$

Above we used the fact that $\mathbb{E}[\mathbb{1}_{z_i=1}] = p_i$ and $\mathbb{E}[\mathbb{1}_{z_i \neq 1}] = 1 - p_i$. This is because the prediction $\hat{y}_i(\mathbf{z})$ is independent of the treatment assignment $z_i$: Crucially, the functions used to learn the prediction for $i$ are not trained on $i$ itself.

We will next analyze the variance of the difference between the estimator and the treatment effect. In order to simplify notation, let $\tau_i = y_i^{(1)} - y_i^{(0)}$ and

$$\hat{\tau}_i(\mathbf{z}) = \frac{y_i^{(1)} - \hat{y}_i(\mathbf{z})}{p_i} \mathbb{1}_{z_i=1} - \frac{y_i^{(0)} - \hat{y}_i(\mathbf{z})}{1-p_i} \mathbb{1}_{z_i \neq 1}.$$

Then we have

$$n^2 \mathrm{Var}[\hat{\tau}(\mathbf{z}) - \tau] = \mathbb{E}\left[ \left( \sum_{i=1}^{n} (\hat{\tau}_i(\mathbf{z}) - \tau_i) \right)^2 \right] = \mathbb{E}\left[ \sum_{i=1}^{n} (\hat{\tau}_i(\mathbf{z}) - \tau_i)^2 \right] + \mathbb{E}\left[ \sum_{i \neq j} (\hat{\tau}_i(\mathbf{z}) - \tau_i)(\hat{\tau}_j(\mathbf{z}) - \tau_j) \right].$$

$$\tag{3}$$

First, we will show that

$$\mathbb{E}\left[ \sum_{i=1}^{n} (\hat{\tau}_i(\mathbf{z}) - \tau_i)^2 \right] = \mathbb{E}\sum_{i=1}^{n} \left( (y_i^{(1)} - \hat{f}_{\mathbf{z},S(i)}^{(1)}(\mathbf{x}_i)) \sqrt{\frac{1-p_i}{p_i}} + (y_i^{(0)} - \hat{f}_{\mathbf{z},S(i)}^{(0)}(\mathbf{x}_i)) \sqrt{\frac{p_i}{1-p_i}} \right)^2.$$

$$\tag{4}$$

In order to simplify the notation, we will use $\hat{y}_i = \hat{y}_i(\mathbf{z})$ when clear from context. We have

$$\mathbb{E}\left[ \sum_{i=1}^{n} (\hat{\tau}_i(\mathbf{z}) - \tau_i)^2 \right] = \sum_{i=1}^{n} \mathbb{E}\left[ (\hat{\tau}_i(\mathbf{z}) - \tau_i)^2 \right] = \sum_{i=1}^{n} \mathbb{E}\left[ \mathbb{E}\left[ (\hat{\tau}_i(\mathbf{z}) - \tau_i)^2 | S_1, S_2, \mathbf{z}_{-\{i\}} \right] \right]. \quad (5)$$

In the first equality, we used linearity of expectation while, in the second equality, we used the law of iterated expectation. Fix $S_1, S_2, z_{-i}$, then

$$(\hat{\tau}_i(\mathbf{z}) - \tau_i)^2 = p_i \left( \frac{y_i^{(1)} - \hat{y}_i}{p_i} - \left( y_i^{(1)} - y_i^{(0)} \right) \right)^2 + (1 - p_i) \left( -\frac{y_i^{(0)} - \hat{y}_i}{1 - p_i} - \left( y_i^{(1)} - y_i^{(0)} \right) \right)^2$$
(6)

$$= p_i \left[ \left( \frac{y_i^{(1)} - \hat{y}_i}{p_i} \right)^2 - 2 \left( \frac{y_i^{(1)} - \hat{y}_i}{p_i} \right) \left( y_i^{(1)} - y_i^{(0)} \right) + \left( y_i^{(1)} - y_i^{(0)} \right)^2 \right]$$

$$+ (1 - p_i) \left[ \left( \frac{y_i^{(0)} - \hat{y}_i}{1 - p_i} \right)^2 + 2 \left( \frac{y_i^{(0)} - \hat{y}_i}{1 - p_i} \right) \left( y_i^{(1)} - y_i^{(0)} \right) + \left( y_i^{(1)} - y_i^{(0)} \right)^2 \right] \quad (7)$$

We foil out each term, divide by $p_i(1 - p_i)$, and group the terms in blue. Then

$$(7) = \frac{1}{p_i(1 - p_i)} \left[ \left( (y_i^{(1)})^2 - 2\hat{y}_i y_i^{(1)} + \hat{y}_i^2 \right)(1 - p_i) - 2 \left( (y_i^{(1)})^2 - \hat{y}_i y_i^{(1)} - y_i^{(1)} y_i^{(0)} + \hat{y}_i y_i^{(0)} \right) p_i(1 - p_i) \right.$$

$$+ \left( (y_i^{(0)})^2 - 2\hat{y}_i y_i^{(0)} + \hat{y}_i^2 \right) p_i + 2 \left( y_i^{(0)} y_i^{(1)} - \hat{y}_i y_i^{(1)} - y_i^{(0)} y_i^{(0)} + \hat{y}_i y_i^{(0)} \right) p_i(1 - p_i)$$

$$\left. + \left( (y_i^{(1)})^2 + -2y_i^{(1)} y_i^{(0)} + (y_i^{(0)})^2 \right)(1 - p_i)p_i \right]$$

$$= \frac{1}{p_i(1 - p_i)} \left[ \left( y_i^{(1)} \right)^2 [1 - p_i - 2p_i(1 - p_i) + (1 - p_i)p_i] \right.$$

$$+ \left( y_i^{(1)} y_i^{(1)} \right) [2p_i(1 - p_i) + 2p_i(1 - p_i) - 2p_i(1 - p_i)]$$

$$+ \left( y_i^{(0)} \right)^2 [p_i + 2p_i(1 - p_i) + (1 - p_i)p_i] + \left( \hat{y}_i y_i^{(1)} \right) [-2(1 - p_i) + 2p_i(1 - p_i) - 2p_i(1 - p_i)]$$

$$\left. + \left( \hat{y}_i y_i^{(0)} \right) [-2p_i(1 - p_i) - 2p_i + 2p_i(1 - p_i)] + (\hat{y}_i)^2(p_i + 1 - p_i) \right] \quad (8)$$

We simplify the factors on the terms in red. Then

$$(8) = \frac{1}{p_i(1 - p_i)} \left[ \left( y_i^{(1)} \right)^2 (1 - p_i)^2 + \left( y_i^{(1)} y_i^{(0)} \right) 2(1 - p_i)p_i + \left( y_i^{(0)} \right)^2 (p_i)^2 \right.$$

$$\left. - \left( \hat{y}_i y_i^{(1)} \right) 2(1 - p_i) - \left( \hat{y}_i y_i^{(0)} \right) 2p_i + (\hat{y}_i)^2 \right]$$

$$= \frac{\left( (1 - p_i)y_i^{(1)} + p_i y_i^{(0)} \right)^2 - 2\hat{y} \left( (1 - p_i)y_i^{(1)} + p_i y_i^{(0)} \right) + \hat{y}^2}{p_i(1 - p_i)}$$

$$= \frac{\left( (1 - p_i)y_i^{(1)} + p_i y_i^{(0)} - \hat{y}_i \right)^2}{p_i(1 - p_i)}$$
(9)

$$= \frac{\left( (1 - p_i)(y_i^{(1)} - \hat{f}_{\mathbf{z}, S(i)}^{(1)}(\mathbf{x}_i)) + p_i(y_i^{(0)} - \hat{f}_{\mathbf{z}, S(i)}^{(0)}(\mathbf{x}_i)) \right)^2}{p_i(1 - p_i)}$$

$$= \left( (y_i^{(1)} - \hat{f}_{\mathbf{z}, S(i)}^{(1)}(\mathbf{x}_i)) \sqrt{\frac{1 - p_i}{p_i}} + (y_i^{(0)} - \hat{f}_{\mathbf{z}, S(i)}^{(0)}(\mathbf{x}_i)) \sqrt{\frac{p_i}{1 - p_i}} \right)^2.$$
(10)

We can check the calculations from Equation 6 to Equation 9 using the WolframAlpha query linked here. The penultimate equality follows from the definition of $\hat{y}_i$. Then plugging 10 into Equation 5 yields Equation 4.

Recall that $\mathbf{z}^{(j \to b)}$ is the treatment assignment vector with $z_j$ set to $b$. Next, we will show that

$$\mathbb{E} \left[ \sum_{i \neq j} (\hat{\tau}_i(\mathbf{z}) - \tau_i)(\hat{\tau}_j(\mathbf{z}) - \tau_j) \right] = \mathbb{E} \left[ \sum_{i \neq j} \left( \hat{y}_i(\mathbf{z}^{(j \to 1)}) - \hat{y}_i(\mathbf{z}^{(j \to 0)}) \right) \left( \hat{y}_j(\mathbf{z}^{(i \to 1)}) - \hat{y}_j(\mathbf{z}^{(i \to 0)}) \right) \right].$$
(11)

For notational brevity, we will use the shorthand

$$\hat{\tau}_i(\mathbf{z}) = \frac{y_i^{(1)} - \hat{y}_i(\mathbf{z})}{p_i}\mathbb{1}_{z_i=1} - \frac{y_i^{(0)} - \hat{y}_i(\mathbf{z})}{1-p_i}\mathbb{1}_{z_i\neq 1} = (-1)^{1-z_i}\frac{y_i^{(z_i)} - \hat{y}_i(\mathbf{z})}{\pi_i}$$

where $\pi_i = p_i\mathbb{1}_{z_i=1} + (1-p_i)\mathbb{1}_{z_i\neq 1}$. We have

$$\mathbb{E}\left[\sum_{i\neq j}(\hat{\tau}_i(\mathbf{z}) - \tau_i)(\hat{\tau}_j(\mathbf{z}) - \tau_j)\right] = \sum_{i\neq j}\mathbb{E}\left[\mathbb{E}\left[(\hat{\tau}_i(\mathbf{z}) - \tau_i)(\hat{\tau}_j(\mathbf{z}) - \tau_j)|S_1, S_2, \mathbf{z}_{-\{i,j\}}\right]\right] \quad (12)$$

where $\mathbf{z}_{-\{i,j\}}$ is the vector $\mathbf{z}$ with the $i$th and $j$th elements removed. The equality follows from linearity of expectation and the law of iterated expectation. We will analyze the conditional expectation

$$\mathbb{E}\left[(\hat{\tau}_i(\mathbf{z}) - \tau_i)(\hat{\tau}_j(\mathbf{z}) - \tau_j)|S_1, S_2, \mathbf{z}_{-\{i,j\}}\right]$$

$$= \mathbb{E}\left[\left((-1)^{1-z_i}\frac{y_i^{(z_i)} - \hat{y}_i(\mathbf{z})}{\pi_i} - \tau_i\right)\left((-1)^{1-z_j}\frac{y_j^{(z_j)} - \hat{y}_j(\mathbf{z})}{\pi_j} - \tau_j\right)\Bigg|S_1, S_2, \mathbf{z}_{-\{i,j\}}\right]$$

$$= \sum_{z_i,z_j\in\{0,1\}}\pi_i\pi_j\left((-1)^{1-z_i}\frac{y_i^{(z_i)} - \hat{y}_i(\mathbf{z}^{(j\to z_j)})}{\pi_i} - \tau_i\right)\left((-1)^{1-z_j}\frac{y_j^{(z_j)} - \hat{y}_j(\mathbf{z}^{(i\to z_i)})}{\pi_j} - \tau_j\right)$$

$$= \sum_{z_i,z_j\in\{0,1\}}\pi_i\pi_j\left((-1)^{1-z_i}(-1)^{1-z_j}\frac{(y_i^{(z_i)} - \hat{y}_i(\mathbf{z}^{(j\to z_j)}))(y_j^{(z_j)} - \hat{y}_j(\mathbf{z}^{(i\to z_i)}))}{\pi_i\pi_j}\right.$$

$$\left.-(-1)^{1-z_i}\frac{y_i^{(z_i)} - \hat{y}_i(\mathbf{z}^{(j\to z_j)})}{\pi_i}\tau_j - (-1)^{1-z_j}\frac{y_j^{(z_j)} - \hat{y}_j(\mathbf{z}^{(i\to z_i)})}{\pi_j}\tau_i + \tau_i\tau_j\right). \quad (13)$$

The first equality follows by the definition of $\hat{\tau}_i$ and $\tau_i$. The second equality follows by expanding the expectation over $z_i$ and $z_j$. The third equality follows by expanding the product of the two terms. We will first analyze the two cross-terms. Without loss of generality, we will analyze the first cross-term. We have

$$\sum_{z_i,z_j\in\{0,1\}}\pi_i\pi_j(-1)^{1-z_i}\frac{y_i^{(z_i)} - \hat{y}_i(\mathbf{z}^{(j\to z_j)})}{\pi_i}\tau_j = \tau_j\sum_{z_i,z_j\in\{0,1\}}\pi_j(-1)^{1-z_i}(y_i^{(z_i)} - \hat{y}_i(\mathbf{z}^{(j\to z_j)}))$$

$$= \tau_j\left(-(1-p_j)(y_i^{(0)} - \hat{y}_i(\mathbf{z}^{(j\to 0)})) - p_j(y_i^{(0)} - \hat{y}_i(\mathbf{z}^{(j\to 1)}))\right. \quad (14)$$

$$\left.+ (1-p_j)(y_i^{(1)} - \hat{y}_i(\mathbf{z}^{(j\to 0)})) + p_j(y_i^{(1)} - \hat{y}_i(\mathbf{z}^{(j\to 1)}))\right)$$

$$= \tau_j\left(p_j\left(y_i^{(1)} - \hat{y}_i(\mathbf{z}^{(j\to 1)}) - y_i^{(0)} + \hat{y}_i(\mathbf{z}^{(j\to 1)})\right)\right. \quad (15)$$

$$\left.+ (1-p_j)\left(y_i^{(1)} - \hat{y}_i(\mathbf{z}^{(j\to 0)}) - y_i^{(0)} + \hat{y}_i(\mathbf{z}^{(j\to 0)})\right)\right)$$

$$= \tau_j\left(p_j(y_i^{(1)} - y_i^{(0)}) + (1-p_j)(y_i^{(1)} - y_i^{(0)})\right) = \tau_j(y_i^{(1)} - y_i^{(0)}) = \tau_j\tau_i. \quad (16)$$

Distributing the sum and plugging in Equation 16 twice, we have

$$(13) = \left(\sum_{z_i,z_j\in\{0,1\}}(-1)^{1-z_i}(-1)^{1-z_j}\left(y_i^{(z_i)} - \hat{y}_i(\mathbf{z}^{(j\to z_j)})\right)\left(y_j^{(z_j)} - \hat{y}_j(\mathbf{z}^{(i\to z_i)})\right)\right) - \tau_i\tau_j.$$

$$(17)$$

Then we have $(13) + \tau_i \tau_j$ equal to

$$
\left( y_i^{(0)} - \hat{y}_i(\mathbf{z}^{(j \to 0)}) \right) \left( y_j^{(0)} - \hat{y}_j(\mathbf{z}^{(i \to 0)}) \right) - \left( y_i^{(0)} - \hat{y}_i(\mathbf{z}^{(j \to 1)}) \right) \left( y_j^{(1)} - \hat{y}_j(\mathbf{z}^{(i \to 0)}) \right)
$$

$$
- \left( y_i^{(1)} - \hat{y}_i(\mathbf{z}^{(j \to 0)}) \right) \left( y_j^{(0)} - \hat{y}_j(\mathbf{z}^{(i \to 1)}) \right) + \left( y_i^{(1)} - \hat{y}_i(\mathbf{z}^{(j \to 1)}) \right) \left( y_j^{(1)} - \hat{y}_j(\mathbf{z}^{(i \to 1)}) \right).
$$

$$
= -\hat{y}_i(\mathbf{z}^{(j \to 0)}) \left( y_j^{(0)} - \hat{y}_j(\mathbf{z}^{(i \to 0)}) - y_j^{(0)} + \hat{y}_j(\mathbf{z}^{(i \to 1)}) \right) + \hat{y}_i(\mathbf{z}^{(j \to 1)}) \left( y_j^{(1)} - \hat{y}_j(\mathbf{z}^{(i \to 0)}) - y_j^{(1)} + \hat{y}_j(\mathbf{z}^{(i \to 1)}) \right)
$$

$$
+ y_i^{(0)} \left( y_j^{(0)} - \hat{y}_j(\mathbf{z}^{(i \to 0)}) - y_j^{(1)} + \hat{y}_j(\mathbf{z}^{(i \to 0)}) \right) - y_i^{(1)} \left( y_j^{(0)} - \hat{y}_j(\mathbf{z}^{(i \to 1)}) - y_j^{(1)} + \hat{y}_j(\mathbf{z}^{(i \to 1)}) \right)
$$

$$
= -\hat{y}_i(\mathbf{z}^{(j \to 0)}) \left( \hat{y}_j(\mathbf{z}^{(i \to 1)}) - \hat{y}_j(\mathbf{z}^{(i \to 0)}) \right) + \hat{y}_i(\mathbf{z}^{(j \to 1)}) \left( \hat{y}_j(\mathbf{z}^{(i \to 1)}) - \hat{y}_j(\mathbf{z}^{(i \to 0)}) \right)
$$

$$
+ y_i^{(0)} \left( y_j^{(0)} - y_j^{(1)} \right) - y_i^{(1)} \left( y_j^{(0)} - y_j^{(1)} \right)
$$

$$
= \left( \hat{y}_i(\mathbf{z}^{(j \to 1)}) - \hat{y}_i(\mathbf{z}^{(j \to 0)}) \right) \left( \hat{y}_j(\mathbf{z}^{(i \to 1)}) - \hat{y}_j(\mathbf{z}^{(i \to 0)}) \right) + \tau_i \tau_j. \tag{18}
$$

Plugging Equation 18 back into Equation 13 and then back into Equation 12 yields Equation 11. Finally, the claimed variance in Equation 3 follows from Equations 4 and 11.

$\square$

# F Connection to Doubly Robust Estimator

The Double-Double estimator described in Algorithm 1 is equivalent to a doubly robust estimator with the same learned functions. We show the algebraic equivalence below.

We used the learned prediction

$$\hat{y}_i(\mathbf{z}) = (1 - p_i)\hat{f}_{\mathbf{z},S(i)}^{(1)}(\mathbf{x}_i) + p_i\hat{f}_{\mathbf{z},S(i)}^{(0)}(\mathbf{x}_i).$$

in the estimator

$$\hat{\tau}(\mathbf{z}) = \frac{1}{n}\sum_{i=1}^{n}\left(\frac{y_i^{(1)} - \hat{y}_i(\mathbf{z})}{p_i}\mathbb{1}_{z_i=1} - \frac{y_i^{(0)} - \hat{y}_i(\mathbf{z})}{1 - p_i}\mathbb{1}_{z_i\neq 1}\right).$$

Plugging in the prediction, the estimator is then

$$\hat{\tau}(\mathbf{z}) = \frac{1}{n}\sum_{i=1}^{n}\left(\frac{y_i^{(1)} - (1 - p_i)\hat{f}_{\mathbf{z},S(i)}^{(1)}(\mathbf{x}_i) - p_i\hat{f}_{\mathbf{z},S(i)}^{(0)}(\mathbf{x}_i)}{p_i}\mathbb{1}_{z_i=1}\right.$$

$$\left. -\frac{y_i^{(0)} - (1 - p_i)\hat{f}_{\mathbf{z},S(i)}^{(1)}(\mathbf{x}_i) - p_i\hat{f}_{\mathbf{z},S(i)}^{(0)}(\mathbf{x}_i)}{1 - p_i}\mathbb{1}_{z_i\neq 1}\right)$$

$$= \frac{1}{n}\sum_{i=1}^{n}\left(\left(\frac{y_i^{(1)} - \hat{f}_{\mathbf{z},S(i)}^{(1)}(\mathbf{x}_i)}{p_i} + \hat{f}_{\mathbf{z},S(i)}^{(1)}(\mathbf{x}_i) - \hat{f}_{\mathbf{z},S(i)}^{(0)}(\mathbf{x}_i)\right)\mathbb{1}_{z_i=1}\right.$$

$$\left. -\left(\frac{y_i^{(0)} - \hat{f}_{\mathbf{z},S(i)}^{(0)}(\mathbf{x}_i)}{1 - p_i} - \hat{f}_{\mathbf{z},S(i)}^{(1)}(\mathbf{x}_i) + \hat{f}_{\mathbf{z},S(i)}^{(0)}(\mathbf{x}_i)\right)\mathbb{1}_{z_i\neq 1}\right)$$

$$= \frac{1}{n}\sum_{i=1}^{n}\left(\frac{y_i^{(1)} - \hat{f}_{\mathbf{z},S(i)}^{(1)}(\mathbf{x}_i)}{p_i}\mathbb{1}_{z_i=1} - \frac{y_i^{(0)} - \hat{f}_{\mathbf{z},S(i)}^{(0)}(\mathbf{x}_i)}{1 - p_i}\mathbb{1}_{z_i\neq 1} + \hat{f}_{\mathbf{z},S(i)}^{(1)}(\mathbf{x}_i) - \hat{f}_{\mathbf{z},S(i)}^{(0)}(\mathbf{x}_i)\right).$$

The final expression is a doubly robust estimator with the same learned functions.

# G Extended Related Work

There are many approaches to treatment effect estimation in the literature. Some of the approaches, like estimators for time series data [BCG17, WSBG18] and design based controls [LD20, Kal18, ADR21, AAM⁺22, HSSZ23], are inappropriate for our setting because we only have one measurement of the outcomes and no control over which observations receive the treatment.

We use propensity scores to account for the probability that an observation received the treatment. There are many estimators that use propensity scores like propensity score matching and propensity stratification [Aus11, Lin14, AS15]. However, in natural experiments, the propensity scores tend to be close to 0 or 1 so propensity score matching and stratification give high variance estimates because of the imbalance in the number of observations that received the treatment or control. Instead, we focus on inverse propensity score weighting and a popular method called the Horvitz-Thompson estimator [HT52, BHAB18]. While the Horvitz-Thompson estimator is similarly prone to high variance, it is common to reduce the variance by adjusting the estimator with a prediction.

There are many estimators that use predictions including regression adjustment, regression discontinuity, and direct regression on the propensity scores [Rho10, CKLP17, CT22]. Some prior work on regression based adjustment estimators tend to make strong assumptions, for example that outcomes are a linear function of the covariates or that the treatment effect is additive [Ros02, TDZL08, NW21a]. For example, regression discontinuity assumes that the treatment effect is not correlated with propensity scores and so can be accurately estimated from observations with similar propensity scores [IL08]. Some work on designing estimators with propensity scores and regression adjustments tend to describe the asymptotic variance at the expense of the constants that effect the performance in the finite setting [Fre08, BLB⁺09, Ken23].

There are several estimators with theoretical guarantees use include propensity scores and learned predictions. Targeted maximum likelihood estimation (TMLE) refines an initial prediction for treatment effect estimation using propensity scores [VdLR⁺11, SR17, Ken23]. Doubly robust estimators are designed to yield asymptotically correct results if they have accurate predictions of either the propensity scores or the outcomes under the treatment and control [SRR99, KS07]. Doubly robust estimators have been extensively studied and optimized in the setting where predictions are linear function [VV15, Tan20].

Recently, there has been substantial work designing neural network architectures and loss functions to estimate treatment effects. DragonNet uses a specialized architecture and targeted regularization [SBV19]. XNet and RANet uses a regression-adjusted pseudo-outcome [KSBY19, CVdS21a]. OffsetNET estimates an offset and TNet uses a vanilla neural network architecture while FlexTENET, SNets, and TARNet use ideas from multi-task and representation learning [CVdS21b]. RNet uses a two-stage optimization approach [NW21b]. PWNet is designed for the Horvitz-Thompson estimator [CVdS21a]. DRNet is designed for a doubly robust estimator [Ken23].

Our work is most similar to two recent papers that have analyzed the Horvitz-Thompson estimator with predictions in the finite population setting. Ghadiri et al. prove theoretical bounds on the variance when the propensity scores are all uniform and the prediction is learned from a linear function [GAM⁺24]. We consider a similar estimator but in the natural experiment setting for general probabilities and with a more powerful prediction learned from nonlinear functions. However, since the functions we use are in general neural networks, we do not give guarantees on the quality of the prediction. The estimator we propose is similar to the Off-policy estimator given by Mou et al. but differs in three ways [MWB22]. First, we learn one function for the treatment outcomes and one for the control outcomes instead of the single function learned by Mou et al. Second, the loss we use to learn the function has an additional multiplicative factor that we theoretically justify. Third, a term in their final estimator has a factor of $\frac{1}{2}$ that we do not have. Together, the differences in our estimator improve its performance substantially over the Mou et al. Off-policy estimator.

# H   Description of Other Estimators

**Regression Discontinuity** The estimator takes the difference between outcomes under the treatment and control in a small region of propensity scores. Let $S_w = \{i : \frac{1}{2} - w \leq p_i \leq \frac{1}{2} + w\}$. The estimator is given by $\text{mean}(\{y^{(1)} : i \in S_w, z_i = 1\}) - \text{mean}(\{y^{(0)} : i \in S_w, z_i \neq 1\})$.

**Propensity Stratification** The estimator takes the average of the difference between mean treatment outcome and mean control outcome over $q$ different $q$-quantiles of the propensity scores. The estimator is given by

$$\frac{1}{q} \sum_{k=1}^{q} \left[ \text{mean}(\{y_i^{(1)} : \frac{k-1}{q} \leq p_i \leq \frac{k}{q}, z_i = 1\}) \right.$$
$$\left. - \text{mean}(\{y_i^{(0)} : \frac{k-1}{q} \leq p_i \leq \frac{k}{q}, z_i \neq 1\}) \right].$$

**Direct Difference** The most naive estimator takes the difference between the outcomes in the treatment group and the outcomes in the control group. The estimator is given by $\frac{2}{n} \sum_{i=1}^{n} \left( y_i^{(1)} \mathbb{1}_{z_i=1} - y_i^{(0)} \mathbb{1}_{z_i \neq 1} \right)$.

**Adjusted Direct** The estimator adjusts the direct estimate by learning a prediction $f(\mathbf{x}) \approx \mathbf{y}^{(1)} \mathbb{1}_{\mathbf{z}=1} + \mathbf{y}^{(0)} \mathbb{1}_{\mathbf{z} \neq 1}$. The estimator is given by $\frac{2}{n} \sum_{i=1}^{n} \left( (y_i^{(1)} - f(\mathbf{x}_i)) \mathbb{1}_{z_i=1} - (y_i^{(0)} - f(\mathbf{x}_i)) \mathbb{1}_{z_i \neq 1} \right)$.

**Horvitz-Thompson** The estimator accounts for the (potentially) non-uniform propensity scores. Horvitz-Thompson estimator is $\frac{1}{n} \sum_{i=1}^{n} \frac{y_i^{(1)}}{p_i} \mathbb{1}_{z_i=1} - \frac{y_i^{(0)}}{1-p_i} \mathbb{1}_{z_i \neq 1}$. Notice that when $p_i = \frac{1}{2}$ for all $i$, the Horvitz-Thompson estimator is equivalent to the direct estimate.

**Targeted Maximum Likelihood Estimator (TMLE)** The TMLE adjusts the learned predictions with an additional regression step. Because of its complexity, we do not describe the full estimator here and instead refer readers to the Step-by-Step Guide in Schuler and Rose [SR17].

**Off-policy** The off-policy estimator due to Mou et al. The off-policy estimator is similar to Double-Double except that estimator learns a single function with a loss weighted by $\frac{1}{\pi_i^2} \mathbb{1}_{z_i=1} + \frac{1}{(1-\pi_i)^2} \mathbb{1}_{z_i \neq 1}$. In addition the final estimator differs by a factor of two on some of the terms.

**Direct Prediction** The estimator takes the difference between *predictions* for the outcomes under the treatment and control. The estimator learns a prediction $f^{(1)}(\mathbf{x}) \approx \mathbf{y}^{(1)}$ and $f^{(0)}(\mathbf{x}) \approx \mathbf{y}^{(0)}$. The estimator is given by $\frac{1}{n} \sum_{i=1}^{n} \left( f^{(1)}(\mathbf{x}_i) - f^{(0)}(\mathbf{x}_i) \right)$.

There are many ways to learn functions for the direct predictions. An extensive line of recent work uses sophisticated neural network architectures and loss functions to account for confounding and other issues. We compare against many of these approaches as implemented in the CATENet benchmark[10] [CVdS21a, CVdS21b].

---

[10] github.com/AliciaCurth/CATENets

# I  Differential Privacy Connection

The second term in the variance described in Theorem 4.1 measures how much changing an observation from the treatment to the control group (and vice versa) affects the adjustment term. Because the second term is 0 for observations in the same partition, notice that the second term would disappear if we only used half the data for estimating (instead of both learning and estimating). In some sense, we can think of the term as the cost of using the same data twice. Since the adjustment term consists of the prediction for the treatment and control outcomes, the second term measures how much removing the observation from the treatment training set and putting it into the control training set (and vice versa) affect the estimators. The quantity is closely related to the requirement of differentially private learning: removing an observation from the training set should not change the learned model too much. Inspired by this connection, we explore whether a popular differentially private learning technique called DP-SGD improves performance [ACG$^+$16, PHK$^+$23]. At each stage of gradient descent, DP-SGD clips the magnitude of the gradient and adds a noise term. From the hyperparameter search in Figure 7, we find that DP-SGD does not improve the estimator. One explanation is that the second term tends to be very small: On the RORCO dataset, we find that the second term is roughly $10^{-30}$.

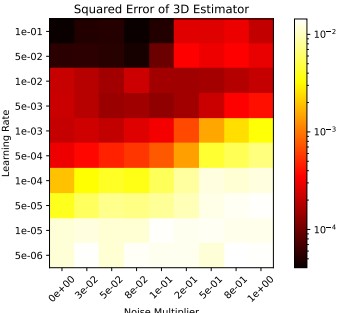

Figure 7: We conduct a hyperparameter search for the Double-Double estimator with differentially private learning. The learning rate controls the step size in gradient descent while the noise multiplier controls the magnitude of the noise added to each gradient. Each square represents the mean squared error on the semi-synthetic data over 100 runs.

Figure 8: Together, the two heatmaps suggest that the Double-Double estimator with differentially private learning achieves lower squared error and is more robust to hyperparameter choices. We use the heatmaps to choose the hyperparameters for the Off-policy + DP and Double-Double + DP estimators in Table **??**.

## J    Benchmark on Additional Datasets

We evaluate the performance of estimators on other datasets. As in our RORCO benchmark, the following table is based on (at least) 100 random runs where the randomness is over the data generation process, propensity score estimation, and any internal randomness in the algorithms. For datasets that do not have both treatment and control outcomes for every observation (ACIC 2016, ACIC 2017, Jobs, and News), we use the synthetic propensity scores and outcomes that we designed for the RORCO semi-synthetic dataset.

| Method | Mean | 1st Quartile | 2nd Quartile | 3rd Quartile | Time (s) |
|---|---|---|---|---|---|
| Regression Discontinuity | 1.46e-03 | 6.33e-04 | 1.09e-03 | 1.95e-03 | 1.05e-03 |
| Propensity Stratification | 1.61e-03 | 1.25e-03 | 1.54e-03 | 1.88e-03 | 2.98e-03 |
| Direct Difference | 4.45e-01 | 3.85e-01 | 4.34e-01 | 5.03e-01 | 4.92e-04 |
| Adjusted Direct | 5.78e-03 | 5.11e-03 | 5.78e-03 | 6.35e-03 | 1.35e+01 |
| Horvitz-Thompson | 5.46e-03 | 7.03e-04 | 3.60e-03 | 7.41e-03 | 4.79e-04 |
| TMLE | 1.42e-01 | 5.11e-03 | 2.81e-02 | 8.59e-02 | 2.69e+01 |
| Off-policy | 3.64e-03 | 2.03e-03 | 3.24e-03 | 4.84e-03 | 1.43e+01 |
| Double-Double | 1.03e-04 | 1.03e-05 | 4.53e-05 | 1.18e-04 | 2.80e+01 |
| Doubly Robust | 1.67e-06 | 1.55e-07 | 6.29e-07 | 2.41e-06 | 2.19e+01 |
| Direct Prediction | 5.08e-03 | 1.37e-03 | 3.69e-03 | 7.63e-03 | 1.38e+01 |
| SNet | 5.67e-02 | 1.53e-02 | 4.99e-02 | 9.58e-02 | 2.39e+01 |
| FlexTENet | 6.65e-04 | 6.07e-05 | 1.78e-04 | 4.16e-04 | 1.50e+02 |
| OffsetNet | 9.26e-04 | 6.43e-04 | 8.80e-04 | 1.07e-03 | 1.37e+02 |
| TNet | 7.59e-04 | 2.05e-05 | 9.02e-05 | 2.66e-04 | 1.21e+02 |
| TARNet | 6.84e-04 | 3.03e-05 | 1.06e-04 | 2.45e-04 | 1.06e+02 |
| DragonNet | 2.07e-02 | 7.07e-03 | 1.83e-02 | 3.22e-02 | 5.66e+00 |
| SNet3 | 3.98e-02 | 7.13e-03 | 2.83e-02 | 6.37e-02 | 1.46e+01 |
| DRNet | 1.41e+01 | 1.32e-03 | 6.56e-03 | 3.61e-02 | 1.31e+02 |
| RANet | 7.63e-04 | 2.53e-05 | 8.49e-05 | 2.48e-04 | 1.95e+02 |
| PWNet | 1.21e+01 | 4.43e-02 | 4.83e-01 | 6.66e+00 | 1.30e+02 |
| RNet | 5.09e-03 | 4.50e-03 | 5.04e-03 | 5.72e-03 | 6.43e+01 |
| XNet | 6.74e-04 | 3.09e-05 | 4.16e-04 | 9.68e-04 | 2.41e+02 |

Table 5: Squared error on the semi-synthetic ACIC 2016 dataset.

| Method | Mean | 1st Quartile | 2nd Quartile | 3rd Quartile | Time (s) |
|---|---|---|---|---|---|
| Regression Discontinuity | 2.77e-03 | 1.69e-03 | 2.30e-03 | 3.58e-03 | 1.01e-03 |
| Propensity Stratification | 1.61e-03 | 1.17e-03 | 1.61e-03 | 1.96e-03 | 2.79e-03 |
| Direct Difference | 4.18e-01 | 3.64e-01 | 4.10e-01 | 4.61e-01 | 4.91e-04 |
| Adjusted Direct | 5.74e-03 | 5.05e-03 | 5.72e-03 | 6.43e-03 | 1.10e+01 |
| Horvitz-Thompson | 5.98e-03 | 8.61e-04 | 3.08e-03 | 7.72e-03 | 4.70e-04 |
| TMLE | 3.47e-01 | 8.57e-03 | 3.63e-02 | 1.74e-01 | 2.33e+01 |
| Off-policy | 4.79e-03 | 2.53e-03 | 3.82e-03 | 6.54e-03 | 2.00e+01 |
| Double-Double | 6.61e-05 | 7.67e-06 | 3.71e-05 | 9.04e-05 | 3.96e+01 |
| Doubly Robust | 1.91e-06 | 1.32e-07 | 6.63e-07 | 2.17e-06 | 1.95e+01 |
| Direct Prediction | 4.23e-03 | 6.91e-04 | 2.73e-03 | 6.54e-03 | 1.28e+01 |
| SNet | 4.79e-02 | 1.12e-02 | 3.88e-02 | 7.13e-02 | 2.02e+01 |
| FlexTENet | 5.37e-04 | 6.12e-05 | 1.78e-04 | 4.01e-04 | 1.48e+02 |
| OffsetNet | 8.82e-04 | 5.67e-04 | 7.68e-04 | 1.10e-03 | 1.32e+02 |
| TNet | 1.42e-03 | 2.97e-05 | 1.45e-04 | 4.59e-04 | 1.14e+02 |
| TARNet | 1.87e-04 | 3.11e-05 | 1.12e-04 | 2.53e-04 | 1.02e+02 |
| DragonNet | 2.17e-02 | 1.02e-02 | 1.77e-02 | 2.94e-02 | 4.35e+00 |
| SNet3 | 3.35e-02 | 5.25e-03 | 1.57e-02 | 5.48e-02 | 1.35e+01 |
| DRNet | 1.80e+02 | 5.76e-04 | 1.83e-03 | 8.69e-03 | 1.20e+02 |
| RANet | 1.42e-03 | 3.56e-05 | 1.41e-04 | 4.02e-04 | 1.84e+02 |
| PWNet | 2.28e+01 | 1.09e-02 | 2.84e-01 | 1.81e+00 | 1.19e+02 |
| RNet | 4.96e-03 | 4.37e-03 | 4.82e-03 | 5.60e-03 | 5.86e+01 |
| XNet | 8.89e-04 | 8.33e-05 | 1.98e-04 | 1.16e-03 | 2.24e+02 |

Table 6: Squared error on the semi-synthetic ACIC 2017 dataset.

| Method | Mean | 1st Quartile | 2nd Quartile | 3rd Quartile | Time (s) |
|---|---|---|---|---|---|
| Regression Discontinuity | 2.26e+00 | 1.54e-01 | 2.40e-01 | 3.35e+00 | 8.37e-04 |
| Propensity Stratification | 1.39e+00 | 8.90e-03 | 2.54e-02 | 2.07e-01 | 1.92e-03 |
| Direct Difference | 4.23e+02 | 3.09e+01 | 6.48e+01 | 1.61e+02 | 4.19e-04 |
| Adjusted Direct | 8.59e+00 | 3.09e+00 | 3.75e+00 | 4.94e+00 | 2.10e+00 |
| Horvitz-Thompson | 3.71e-01 | 1.72e-02 | 4.81e-02 | 2.13e-01 | 3.85e-04 |
| TMLE | 5.22e+00 | 6.73e-02 | 3.13e-01 | 1.60e+00 | 3.98e+00 |
| Off-policy | 5.44e-01 | 3.23e-01 | 4.75e-01 | 6.91e-01 | 2.01e+00 |
| Double-Double | 2.01e-01 | 3.29e-02 | 1.15e-01 | 3.02e-01 | 4.00e+00 |
| Doubly Robust | 7.67e-02 | 2.17e-03 | 5.59e-03 | 2.43e-02 | 3.32e+00 |
| Direct Prediction | 1.86e+00 | 1.10e-02 | 3.97e-02 | 2.12e-01 | 2.08e+00 |
| FlexTENet | 1.03e+01 | 1.72e-02 | 5.91e-01 | 1.24e+00 | 1.03e+01 |
| OffsetNet | 3.64e+00 | 5.66e-02 | 1.91e-01 | 7.04e-01 | 3.95e+00 |
| TNet | 4.38e-01 | 4.00e-02 | 3.34e-01 | 5.98e-01 | 4.51e+00 |
| TARNet | 5.16e+00 | 1.04e-02 | 2.46e-01 | 8.00e-01 | 3.44e+00 |
| SNet3 | 3.77e+00 | 4.46e-01 | 7.55e-01 | 1.09e+00 | 1.33e+01 |
| DRNet | 9.79e-01 | 5.71e-02 | 9.00e-02 | 4.38e-01 | 8.99e+00 |
| RANet | 3.43e-01 | 3.90e-02 | 1.47e-01 | 5.80e-01 | 7.58e+00 |
| PWNet | 9.61e+00 | 4.85e-02 | 5.57e-01 | 1.24e+00 | 8.82e+00 |
| RNet | 1.66e+00 | 2.83e-02 | 2.08e-01 | 1.17e+00 | 6.50e+00 |
| XNet | 6.13e-01 | 1.44e-02 | 9.30e-02 | 2.35e-01 | 1.10e+01 |

Table 7: Squared error on the IHDP dataset.

| Method | Mean | 1st Quartile | 2nd Quartile | 3rd Quartile | Time (s) |
|---|---|---|---|---|---|
| Regression Discontinuity | 3.27e-03 | 1.47e-03 | 2.63e-03 | 4.27e-03 | 8.39e-04 |
| Propensity Stratification | 9.73e-04 | 4.03e-04 | 6.05e-04 | 1.06e-03 | 1.88e-03 |
| Direct Difference | 1.84e-01 | 1.11e-01 | 1.70e-01 | 2.26e-01 | 4.14e-04 |
| Adjusted Direct | 2.17e-03 | 1.17e-03 | 1.92e-03 | 2.59e-03 | 1.59e+00 |
| Horvitz-Thompson | 5.85e-03 | 3.42e-04 | 1.71e-03 | 7.87e-03 | 3.81e-04 |
| TMLE | 3.43e-03 | 1.19e-04 | 7.15e-04 | 3.49e-03 | 3.40e+00 |
| Off-policy | 1.15e-03 | 3.94e-04 | 6.85e-04 | 1.04e-03 | 1.84e+00 |
| Double-Double | 1.40e-04 | 6.04e-06 | 4.01e-05 | 1.43e-04 | 3.56e+00 |
| Doubly Robust | 3.06e-05 | 8.12e-07 | 4.32e-06 | 1.41e-05 | 2.92e+00 |
| Direct Prediction | 1.55e-03 | 9.36e-05 | 4.43e-04 | 1.51e-03 | 1.80e+00 |
| SNet | 4.88e-03 | 4.17e-04 | 1.80e-03 | 5.01e-03 | 2.96e+01 |
| FlexTENet | 2.25e-03 | 5.90e-05 | 3.35e-04 | 1.58e-03 | 8.35e+01 |
| OffsetNet | 2.63e-04 | 4.93e-05 | 1.52e-04 | 3.50e-04 | 6.23e+01 |
| TNet | 3.23e-03 | 2.04e-04 | 8.16e-04 | 2.90e-03 | 4.63e+01 |
| TARNet | 2.05e-03 | 1.09e-04 | 4.18e-04 | 1.57e-03 | 4.84e+01 |
| DragonNet | 7.78e-03 | 5.25e-04 | 2.44e-03 | 7.85e-03 | 2.55e+00 |
| SNet3 | 4.98e-03 | 5.24e-04 | 1.91e-03 | 5.25e-03 | 2.14e+01 |
| DRNet | 2.94e-03 | 7.18e-05 | 4.92e-04 | 2.42e-03 | 5.64e+01 |
| RANet | 3.25e-03 | 2.16e-04 | 9.16e-04 | 2.96e-03 | 8.09e+01 |
| PWNet | 8.08e-03 | 9.05e-04 | 3.59e-03 | 9.82e-03 | 4.93e+01 |
| RNet | 5.68e-04 | 2.52e-04 | 4.76e-04 | 7.19e-04 | 2.73e+01 |
| XNet | 5.99e-04 | 1.17e-05 | 6.61e-05 | 3.31e-04 | 1.01e+02 |

Table 8: Squared error on the semi-synthetic JOBS dataset.

| Method | Mean | 1st Quartile | 2nd Quartile | 3rd Quartile | Time (s) |
|---|---|---|---|---|---|
| Regression Discontinuity | 2.28e-03 | 1.23e-03 | 2.09e-03 | 3.25e-03 | 1.16e-03 |
| Propensity Stratification | 4.96e-04 | 3.11e-04 | 5.02e-04 | 6.15e-04 | 2.90e-03 |
| Direct Difference | 8.14e-02 | 3.39e-02 | 7.18e-02 | 1.27e-01 | 4.86e-04 |
| Adjusted Direct | 6.00e-04 | 2.14e-04 | 4.78e-04 | 8.84e-04 | 1.20e+01 |
| Horvitz-Thompson | 7.31e-04 | 7.56e-05 | 2.73e-04 | 8.64e-04 | 4.79e-04 |
| TMLE | 2.48e-03 | 5.72e-06 | 3.21e-05 | 1.93e-04 | 2.49e+01 |
| Off-policy | 5.30e-04 | 2.51e-04 | 4.77e-04 | 7.28e-04 | 1.27e+01 |
| Double-Double | 1.33e-07 | 2.91e-09 | 1.84e-08 | 8.87e-08 | 2.54e+01 |
| Doubly Robust | 3.68e-08 | 9.92e-10 | 9.88e-09 | 4.63e-08 | 2.04e+01 |
| Direct Prediction | 1.30e-05 | 1.13e-06 | 4.41e-06 | 1.64e-05 | 1.24e+01 |
| SNet | 2.27e-04 | 2.85e-05 | 1.36e-04 | 3.19e-04 | 7.60e+01 |
| FlexTENet | 2.92e-05 | 4.28e-06 | 1.36e-05 | 2.89e-05 | 1.82e+02 |
| OffsetNet | 2.66e-05 | 2.43e-06 | 1.02e-05 | 3.05e-05 | 1.33e+02 |
| TNet | 2.26e-05 | 2.24e-06 | 1.18e-05 | 2.69e-05 | 1.26e+02 |
| TARNet | 2.73e-05 | 1.23e-06 | 1.14e-05 | 3.26e-05 | 9.01e+01 |
| DragonNet | 4.90e-05 | 2.26e-06 | 2.43e-05 | 5.97e-05 | 3.83e+01 |
| SNet3 | 3.88e-04 | 3.90e-05 | 1.88e-04 | 4.79e-04 | 6.14e+01 |
| DRNet | 2.83e-05 | 1.40e-06 | 8.82e-06 | 1.91e-05 | 1.90e+02 |
| RANet | 2.16e-05 | 3.34e-06 | 1.10e-05 | 3.26e-05 | 1.91e+02 |
| PWNet | 6.40e-04 | 4.49e-05 | 2.93e-04 | 8.10e-04 | 1.45e+02 |
| RNet | 4.37e-05 | 5.20e-06 | 2.13e-05 | 6.29e-05 | 6.31e+01 |
| XNet | 5.69e-06 | 2.70e-07 | 1.02e-06 | 5.90e-06 | 2.41e+02 |

Table 9: Squared error on the semi-synthetic NEWS dataset.

| Method | Mean | 1st Quartile | 2nd Quartile | 3rd Quartile | Time (s) |
|---|---|---|---|---|---|
| Regression Discontinuity | 4.27e-05 | 2.40e-05 | 3.84e-05 | 6.08e-05 | 1.11e-03 |
| Propensity Stratification | 3.28e-05 | 3.53e-06 | 1.44e-05 | 5.11e-05 | 3.09e-03 |
| Direct Difference | 2.45e-05 | 2.56e-06 | 1.02e-05 | 3.66e-05 | 4.86e-04 |
| Adjusted Direct | 1.73e-01 | 1.25e-03 | 4.45e-03 | 2.79e-02 | 1.22e+01 |
| Horvitz-Thompson | 8.52e-05 | 8.84e-06 | 3.95e-05 | 9.34e-05 | 4.65e-04 |
| TMLE | 2.66e+00 | 2.60e-02 | 1.05e-01 | 2.87e-01 | 2.48e+01 |
| Off-policy | 8.65e-03 | 6.14e-04 | 2.60e-03 | 8.86e-03 | 1.24e+01 |
| Double-Double | 9.95e-03 | 5.25e-04 | 2.69e-03 | 1.01e-02 | 2.46e+01 |
| Doubly Robust | 1.89e-02 | 2.20e-04 | 1.30e-03 | 5.40e-03 | 2.10e+01 |
| Direct Prediction | 1.53e-01 | 2.01e-02 | 8.36e-02 | 2.24e-01 | 1.26e+01 |
| FlexTENet | 9.36e+01 | 1.80e-01 | 1.68e+00 | 1.22e+01 | 2.04e+01 |
| OffsetNet | 1.19e+00 | 2.78e-02 | 8.62e-02 | 5.83e-01 | 1.61e+01 |
| TNet | 2.16e+01 | 3.08e-02 | 2.49e-01 | 9.05e-01 | 1.01e+01 |
| TARNet | 4.06e+00 | 4.86e-02 | 1.66e-01 | 8.02e-01 | 9.54e+00 |
| RANet | 1.04e+01 | 1.02e-01 | 5.73e-01 | 4.25e+00 | 1.72e+01 |

Table 10: Squared error on the TWINS dataset.

# K   Dataset Summary

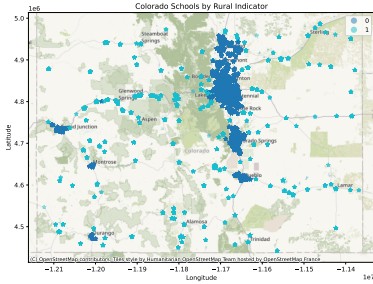

Figure 9: A map of schools in Colorado. The color indicates whether a school is "rural". At the suggestion of RORCO, we only consider rural schools because, in rural areas, it is a more reasonable assumption that students attend the school closest to their medical provider.

Figure 10: A map of rural K-12 public schools in Colorado. The color indicates whether each grade at the associated school "received" the RORCO treatment. We determine that a grade received the treatment if nearby RORCO clinics gave books to more than half the number of students in the grade.

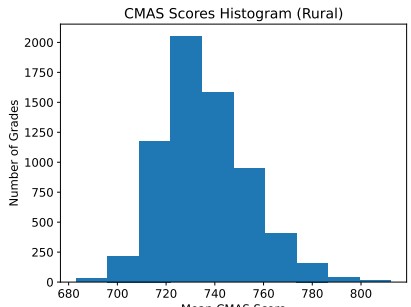

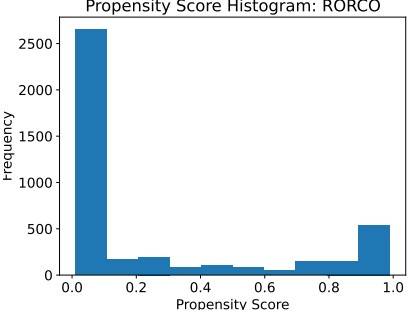

Figure 11: Histogram of CMAS scores by grade for rural schools.

Figure 12: Histogram of propensity scores. Since the dataset is imbalanced (roughly one quarter of observations receive the treatment), the propensities are skewed to 0.

Table 11: A summary of covariates in the RORCO dataset.

| | count | mean | std | min | 50% | max |
|---|---|---|---|---|---|---|
| Low Grade | 4178 | 28.3492 | 29.4807 | 2 | 6 | 90 |
| High Grade | 4178 | 73.2312 | 24.9566 | 20 | 80 | 120 |
| Latitude | 4178 | 39.0379 | 1.02844 | 37.0191 | 39.2469 | 40.8236 |
| Longitude | 4178 | -105.645 | 1.64942 | -108.904 | -105.52 | -102.123 |
| County Code | 4178 | 33.529 | 18.9398 | 1 | 32 | 98 |
| District Code | 4178 | 1756.8 | 1150.84 | 50 | 1500 | 8001 |
| K-12 Count | 4178 | 284.362 | 148.374 | 25 | 259.5 | 1132 |
| Free Lunch | 4178 | 92.7475 | 74.8659 | 0 | 75 | 418 |
| Reduced Lunch | 4178 | 24.432 | 18.3776 | 0 | 21 | 139 |
| Paid Lunch | 4178 | 146.765 | 111.575 | 0 | 128 | 816 |
| Free And Reduced Count | 4178 | 117.18 | 87.8908 | 0 | 99 | 496 |
| % Free | 4178 | 0.330949 | 0.191124 | 0 | 0.32 | 0.831 |
| % Reduced | 4178 | 0.0887836 | 0.0487366 | 0 | 0.09 | 0.253 |
| % Free And Reduced | 4178 | 0.419733 | 0.218246 | 0 | 0.42 | 0.908 |
| Pk-12 Pupil Membership | 4178 | 295.502 | 149.433 | 25 | 271 | 1132 |
| Sped Count | 4178 | 39.2197 | 23.048 | 0 | 35 | 130 |
| Sped Pct | 4178 | 0.132976 | 0.0442545 | 0 | 0.131 | 0.316 |
| El Count | 4178 | 34.6029 | 51.1494 | 0 | 12 | 234 |
| El Pct | 4178 | 0.0994809 | 0.131769 | 0 | 0.042 | 0.764 |
| Homeless Count | 4178 | 3.51843 | 6.58953 | 0 | 0 | 52 |
| Homeless Pct | 4178 | 0.01232 | 0.0285948 | 0 | 0 | 0.254 |
| Gifted And Talented Count | 4178 | 13.4165 | 19.9115 | 0 | 7 | 193 |
| Gt Pct | 4178 | 0.0411024 | 0.0424374 | 0 | 0.031 | 0.275 |
| Online Count | 4178 | 2.98923 | 29.0468 | 0 | 0 | 373 |
| Online Pct | 4178 | 0.0128119 | 0.109844 | 0 | 0 | 1 |
| Section 504 Count | 4178 | 5.07157 | 8.05086 | 0 | 0 | 95 |
| Section 504 Pct | 4178 | 0.0161197 | 0.0217525 | 0 | 0 | 0.125 |
| Immigrant Count | 4178 | 2.53614 | 7.83388 | 0 | 0 | 71 |
| Immigrant Pct | 4178 | 0.00640522 | 0.0184535 | 0 | 0 | 0.158 |
| Migrant Count | 4178 | 0.83102 | 3.05824 | 0 | 0 | 22 |
| Migrant Pct | 4178 | 0.00275108 | 0.0106982 | 0 | 0 | 0.082 |
| Distr Code | 4178 | 1756.8 | 1150.84 | 50 | 1500 | 8001 |
| Pre-K | 4178 | 11.14 | 19.1539 | 0 | 0 | 108 |
| Half-Day K | 4178 | 0.0143609 | 0.168886 | 0 | 0 | 3 |
| Full-Day K | 4178 | 22.0682 | 24.3982 | 0 | 17 | 98 |
| 2019-2020 students Counted | 4178 | 322.656 | 170.878 | 1 | 299 | 1202 |
| Days In Session Reported | 4178 | 141.349 | 25.7376 | 15 | 145 | 230 |
| Attendance Rate* | 4178 | 0.940132 | 0.0597037 | 0 | 0.946 | 1.181 |
| Truancy Rate** | 4178 | 0.0130524 | 0.017604 | 0 | 0.01 | 0.415 |
| Days Attended | 4178 | 41074.1 | 23406.6 | 0 | 37366.1 | 179937 |
| student Days Excused Absence | 4178 | 1955 | 1811.02 | 0 | 1740.25 | 23719.8 |
| student Days Unexcused Absence | 4178 | 565.83 | 650.018 | 0 | 382.5 | 7129.1 |
| Days Possible Attendance | 4178 | 43591.3 | 24766.8 | 15 | 39675 | 187764 |
| County Code | 4178 | 33.529 | 18.9398 | 1 | 32 | 98 |
| Teacher FTE | 4178 | 20.5824 | 9.55585 | 2.1 | 19.4 | 67.7 |
| District Code | 4178 | 1804.14 | 1261.19 | 50 | 1510 | 8001 |
| FTE | 4178 | 51.6705 | 183.902 | 0 | 0 | 1130.07 |
| Average Salary | 4178 | 22468.9 | 27367.3 | 0 | 0 | 78568 |
| FTE | 4178 | 288.719 | 689.855 | 0 | 86.2612 | 4053.69 |
| Average Salary | 4178 | 51776.5 | 11894.6 | 0 | 50639 | 88981 |
| Num_From_Rorco | 4178 | 15.8511 | 29.9759 | 0 | 0 | 237 |
| Capacity | 4178 | 58.2461 | 45.6389 | 0 | 45 | 306 |
| Half_Rorco | 4178 | 0.229536 | 0.420585 | 0 | 0 | 1 |
| Nearby_students | 4178 | 3651.82 | 2726.3 | 7 | 3290 | 9949 |
| Is_Rural | 4178 | 1 | 0 | 1 | 1 | 1 |

# L    Other Learning Models

There are many models that we could use to learn the outcomes. We do explore different models i.e., the 12 CATENet estimators in our benchmark each use their own custom model. For the remaining structures, we default to a shallow neural network. We used the same neural network so that we can evaluate the estimators on a level playing field, focusing on the way the estimates are created rather than the power of the model.

To explore the role of different learning models, we ran experiments with BART and a causal forest. The causal forest model takes as input the covariates, treatment assignment, and observed outcomes so we implement this as its own estimator. For each model that supports generic models (i.e., every method except the causal forest and the CATENet estimators), we ran the model with the BART model and default parameters. The below table summarizes the results from 10 runs on the RORCO dataset.

Across the board, the performance is worse with BART than with the shallow neural network. In fact, the propensity stratification method that does not use the learned predictions generally gives the third best performance. Besides this, the relative performance between estimators is generally preserved.

| Method | Mean | 1st Quartile | 2nd Quartile | 3rd Quartile | Time (s) |
|---|---|---|---|---|---|
| Regression Discontinuity | 4.40e-03 | 3.06e-03 | 3.77e-03 | 6.18e-03 | 3.55e-04 |
| Propensity Stratification | 2.72e-03 | 2.09e-03 | 2.49e-03 | 3.63e-03 | 1.90e-03 |
| Direct Difference | 4.81e-01 | 3.79e-01 | 4.12e-01 | 5.83e-01 | 1.48e-04 |
| Adjusted Direct | 5.11e-02 | 4.36e-02 | 5.21e-02 | 5.68e-02 | 1.84e+01 |
| Horvitz-Thompson | 9.13e-03 | 1.94e-03 | 6.98e-03 | 1.19e-02 | 2.37e-04 |
| TMLE | 1.35e-01 | 3.48e-02 | 7.49e-02 | 1.93e-01 | 4.53e+01 |
| Off-policy | 6.89e-03 | 4.05e-03 | 6.43e-03 | 9.80e-03 | 3.17e+01 |
| Double-Double | 1.30e-03 | 8.98e-04 | 1.04e-03 | 1.57e-03 | 4.73e+01 |
| Doubly Robust | 1.17e-03 | 8.16e-04 | 1.13e-03 | 1.48e-03 | 3.64e+01 |
| Direct Prediction | 1.81e-01 | 1.51e-01 | 1.86e-01 | 1.97e-01 | 2.77e+01 |
| Causal Forest | 3.02e-01 | 2.76e-01 | 3.05e-01 | 3.27e-01 | 9.30e-01 |

Table 12: Squared error on the semi-synthetic RORCO dataset with the BART learning model. The summary statistics are computed over 10 runs. The randomness in the runs comes from the synthetically generated outcomes, estimates of the propensity scores, and any internal randomness in the estimators. Note that we adopt the Olympic medal convention: gold , silver and bronze cell highlights signify first, second and third best performance, respectively.

