# OpenReview forum: "Benchmarking Estimators for Natural Experiments: A Novel Dataset and a Doubly Robust Algorithm"
_NeurIPS.cc/2024/Datasets_and_Benchmarks_Track — NeurIPS 2024 Track Datasets and Benchmarks Poster_

### Official Review · Reviewer_CMHp · 2024-07-08
**Important topic, more details needed.**

**Rating:** 6
**Confidence:** 4
**Clarity:** The paper is clearly written but some…

**Review:**

Overall, I believe this paper studies an important problem and presents unique datasets that could potentially be useful to the community. I also appreciate the contributions in benchmarking many estimators. However, the clarity -- including details for constructing synthetic data, implicit assumptions in the data generating processes, details for propensity score estimation, and so on -- needs improvement. Also, I am not fully convinced about the necessity/usefulness of the theoretical analysis.

**Strengths:**

1. This paper studies an important problem and provides good resources for the community to use.

2. The empirical experiments in this paper are fruitful and useful as future reference.

3. This paper has positive social implications.

**Additional Feedback:**

Questions for the authors:

1. What are the quartiles in Table 4 over? It seems that there is no synthetic data generating process so each estimator should just return one estimate from one data?

2. In Figure 1, how are the propensity scores estimated? I was wondering how accurate they are and how much we can trust the estimation.

3. It would be important to describe how the propensity scores are estimated for Figure 3.

4. In Section 3.3, it is not very clear how the experiments are operated. Are the propensity scores first estimated in some way and then perturbed with random noise? In practice, the accuracy may be driven by the estimation methods, which can be quite different from adding noise to decrease accuracy. I would suggest the authors justify this approach, or add experiments that vary the estimation method for the propensity score and investigate the relationship between the estimation accuracy of propensity score and treatment effect.

5. It is a bit confusing why the regression function is considered estimated but the propensity score is assumed correct in Theorem 4.1.

6. One important aspect in people's preference for doubly robust estimators is that they enable valid $\sqrt{n}$-normal inference even if the predictors converge at slow rates. Perhaps investigating the resulting inference would also be an important aspect in a benchmark.

7. The messages from Section 4 should be made more clear. Does the analysis lead to practical recommendations? It seems a bit unsatisfying that the theoretically inspired method does not beat the non-splitting doubly robust estimator.

8. Why do you decide to use the same fixed prediction structure (neural network) for all estimators for evaluation? For instance, how one may fit the prediction models is also an active topic in the literature, and models such as BART and causal forests are also widely used.

9. Besides better documentation and easier access, in terms of the datasets themselves, do you think these datasets are more interesting than existing ones?

Minor points:

1. It would be helpful to move the captions in Figure 4 a bit in order not to overlap with the curves.

2. Line 237, the "." before "A" should be a ",".

**Correctness:**

I would suggest the authors include more details about the construction of the data and the benchmarking methods to help me better judge the correctness of the submission. Please see Additional Feedback section.

**Documentation:**

I would suggest the authors to improve documentation by adding details on how the datasets are created, etc.

**Ethics:**

There might be privacy issue but I would trust the authors (they mentioned privacy in the paper) that this is not concerning.

**Limitations:**

Yes.

**Opportunities For Improvement:**

1. The details and implicit assumptions in the synthetic data generating processes should be more clearly stated.

2. The emphasis on the regression component but not the propensity score component is a bit confusing.

3. Using the same fixed prediction structure (neural network) for all estimators may be inadequate for evaluation. For instance, how one may fit the prediction models is also an active topic in the literature, and models such as BART and causal forests are also widely used.

4. How the propensity scores are estimated should be described in more detail.

Please see the Additional Feedback section for more specific questions.

**Relation To Prior Work:**

I understand that this work is motivated by shortcomings of other datasets/benchmarks in causal inference, including lack of documentation, difficulty to access the datasets, etc. However, this work also needs more work in terms of documentation, and more details are needed as I commented in other blocks.

**Summary And Contributions:**

Motivated by the importance of treatment effect estimation and challenges in benchmarking and assessing related datasets, this paper creates datasets from the RORCO organization and benchmarks 20+ estimators for estimating treatment effects using synthetic data. The experiments suggest the superior performance of doubly robust estimators. Motivated by this, the authors also provide some analysis on the variance of the doubly robust estimator, based on which they propose a new estimator that performs best among competitors with sample splitting but underperforms the doubly robust estimator.

---

> ### Author Rebuttal · Authors · 2024-08-16
>
> Dear Reviewer,
>
> Thank you for your time and feedback! We will address your concerns here and add the relevant details to the revised version of the paper.
>
> **Synthetic Data Generation**
>
> The literacy experts we talked to made the following suggestions for the synthetic outcomes:
>
> (1) The outcomes should be inversely related to the propensity. That is, students are more likely to participate in the literacy program if they have lower literacy proficiency.
>
> (2) The treatment effect should be negligible for small propensity scores and increasing for large propensity scores. That is, students who are less likely to participate in the program, and so have higher literacy proficiency as per (1), will not benefit from the program because they already have sufficient resources. In contrast, students who are more likely to participate in the program, and so have lower literacy proficiency as per (1), will benefit from the program in proportion to their literacy needs.
>
> Based on (1) and (2), we made the control outcomes vary between 0 and 1 with an inverse linear relationship to propensity scores. (We generated the propensity scores by creating a random linear combination of the covariates.) We made the treatment outcomes align with the control outcomes until .5 and then increasingly separate (with a negated square root added to the control outcomes). We have expanded on the details for the revised version of the paper; the code is [here](https://github.com/rtealwitter/naturalexperiments/blob/b3a3e71597788b9c4451809da14ec1f87fd17df4/naturalexperiments/utils.py#L48) for reference.
>
> **Regression vs Propensity Error**
>
> As you point out, our theoretical analysis assumes that the propensity scores are known exactly. We make this assumption because it simplifies the analysis (which is already quite involved in the finite setting, i.e., see Appendix D). Understanding how robust the variance analysis is to propensity score accuracy is an important direction for future work.
>
> **Learning Models**
>
> There are many models that we could use to learn the outcomes. We do explore different models i.e., the 12 CATENet estimators in our benchmark each use their own custom model. For the remaining structures, you are correct that we default to a shallow neural network. We used the same neural network so that we can evaluate the estimators on a level playing field, focusing on the way the estimates are created rather than the power of the model.
>
> At your suggestion, we ran experiments with BART and a causal forest. The causal forest model takes as input the covariates, treatment assignment, and observed outcomes so we implement this as its own estimator. For each model that supports generic models (i.e., every method except the causal forest and the CATENet estimators), we ran the model with the BART model and default parameters. The below table summarizes the results from 10 runs on the RORCO dataset.
>
> | Method                    |    Mean |   1st Quartile |   2nd Quartile |   3rd Quartile |   Time (s) |
> |---------------------------|---------|----------------|----------------|----------------|------------|
> | Regression Discontinuity  | 0.0044  |       0.00306  |        0.00377 |        0.00618 |   0.000355 |
> | Propensity Stratification | 0.00272 |       0.00209  |        0.00249 |        0.00363 |   0.0019   |
> | Direct Difference         | 0.481   |       0.379    |        0.412   |        0.583   |   0.000148 |
> | Adjusted Direct           | 0.0511  |       0.0436   |        0.0521  |        0.0568  |  18.4      |
> | Horvitz-Thompson          | 0.00913 |       0.00194  |        0.00698 |        0.0119  |   0.000237 |
> | TMLE                      | 0.135   |       0.0348   |        0.0749  |        0.193   |  45.3      |
> | Off-policy                | 0.00689 |       0.00405  |        0.00643 |        0.0098  |  31.7      |
> | Double-Double             | 0.0013  |       0.000898 |        0.00104 |        0.00157 |  47.3      |
> | Doubly Robust             | 0.00117 |       0.000816 |        0.00113 |        0.00148 |  36.4      |
> | Direct Prediction         | 0.181   |       0.151    |        0.186   |        0.197   |  27.7      |
> | Causal Forest             | 0.302   |       0.276    |        0.305   |        0.327   |   0.93     |
>
> The doubly robust estimators still give the best performance but, likely because it is more expressive, the shallow neural network (results in the current version of the paper) gives substantially better results for all the estimators.
>
> **Propensity Scores**
>
> We estimate the propensity scores by training a shallow neural network on the treatment assignment and covariates with binary cross entropy loss. We regularize the predicted propensities by clipping them to the range between .01 and .99. The code for this process is available [here](https://github.com/rtealwitter/naturalexperiments/blob/b3a3e71597788b9c4451809da14ec1f87fd17df4/naturalexperiments/model.py#L93). (We also tried using a scikit-learn implementation of logistic regression and found similar performance. But, to reduce the complexity of the package dependencies, we opted to use a custom implementation.) We have expanded on these details in the revised version of the paper.
>
> **Documentation of Dataset Construction**
>
> In order to provide more documentation, we have created [this](https://github.com/rtealwitter/naturalexperiments/blob/main/naturalexperiments/data/rorco/rorco_documentation.md) detailed description of the dataset construction process and made it accessible in the repo.
>
> Due to space constraints, we respond to your questions in a comment below.

---

> > ### Comment · Reviewer_CMHp · 2024-08-25
> >
> > Thank you for the detailed and fruitful response to my comments! I raise my score to 6.

---

> ### Author Response · Authors · 2024-08-16
> **Answers to Questions**
>
> 1. The quartiles in Table 4 are over the following sources of randomness: The propensity scores are generated from a learning process that uses random batches for training, the learned predictions (in most of the estimators) are also generated from a learning process that uses batches, and, in some estimators like Double-Double, there is a random training-testing split.
>
> 2. The propensity scores in Figure 1 (and the rest of the paper except for Figure 6) are generated from a shallow neural network with a cross entropy loss and then clipped to the range between .01 and .99. As Figure 3 suggests, the propensities are well calibrated (close to the identity line). In additional experiments on RORCO that we just ran (code [here](https://github.com/rtealwitter/naturalexperiments/blob/main/paper_experiments/cross_entropy.py)), we find the cross entropy between the true propensities and a sampled treatment assignment is $.202 \pm .029$ while the cross entropy between the predicted propensities and a sampled treatment assignment vector is $.196 \pm .029$ where the standard deviation is over 100 synthetic generations of the propensity scores. This suggests that the predicted propensity scores are remarkably accurate, at least on the synthetic data where we know the true propensity scores.
>
> 3. Please see the answer to the prior question.
>
> 4. The goal of the experiments in Section 3.3 is to investigate the accuracy of the predictions as the accuracy of the predicted propensity scores vary. To accomplish this, we use the true propensity scores (from the synthetic experiments) and add varying levels of random noise. We then plot the performance by the cross entropy between the treatment assignment and the noised propensity scores. We acknowledge that it would be interesting to explore other sources of error in the propensity scores besides adding random noise. For example, we could use models of varying complexity to estimate the propensity from synthetic data. This is a good suggestion, we will consider adding such experiments to the revised paper.
>
> 5. Please see our response under **Regression vs Propensity Error** in our main rebuttal.
>
> 6. It is an excellent suggestion to explore asymptotic inference in the benchmark. We could accomplish this by creating a large synthetic data set, and we will consider how to do so in the future version of the paper. Right now, we are limited by the size of the RORCO dataset since only the outcomes (not the covariates) are synthetic.
>
> 7. In Section 4, we theoretically analyze doubly robust estimators with a testing-training split and motivate a new method. The message is that doubly-robust estimators like the one we suggest are useful in practice. While perhaps slightly unsatisfying that the non-splitting method performs better, this is not surprising, as it effectively has access to twice the data. Splitting is required for the theoretical variance bound, but not for the correctness of the estimator. So we think the analysis still justifies why the non-splitting method does well. A natural question for future work would be a full analysis of the non-splitting method. We have made this message more clear in the revised version.
>
> 8. Please see our response above under **Learning Models** in our main rebuttal.
>
> 9. We do think our dataset is more interesting! In the context of the existing datasets (please see the detailed discussion under our **Qualitative Dataset Evaluation** response to Reviewer ozjC), we think the community should A) use data from real natural experiments and B) collaborate with domain experts when synthetic outcomes are needed. Further, the outcomes suggested by the literacy experts for RORCO are quite interesting and non-standard: the outcomes and the treatment effect are correlated with propensity scores. This setting reflects natural experiments conducted by nonprofits which (we believe) is an excellent high-impact area for future academic research.
>
> **Minor Points**
>
> Thank you for your minor points of feedback. We have corrected the typo and you can find the updated figures with smaller legends [here](https://github.com/rtealwitter/naturalexperiments/blob/main/paper_experiments/images/RORCO%3A%20Squared%20Error%20by%20Number%20of%20Observations.pdf), [here](https://github.com/rtealwitter/naturalexperiments/blob/main/paper_experiments/images/RORCO%3A%20Squared%20Error%20by%20Cross%20Entropy.pdf), and [here](https://github.com/rtealwitter/naturalexperiments/blob/main/paper_experiments/images/RORCO%3A%20Squared%20Error%20by%20Correlation.pdf).

---

### Official Review · Reviewer_ozjC · 2024-07-24

**Rating:** 7
**Confidence:** 3
**Correctness:** na.
**Clarity:** very well written paper.

**Review:**

I like this paper. While I have some reservations about including a new method in a benchmark track paper, it seems the introduction of a new dataset with some added benchmarks on top of it is the ideal candidate for this track. Furthermore, I agree with the authors that the treatment effects community is in dire need of new (more modern) datasets to test novel algorithms. While TWINS seems to have more samples, RORCO does look like it has more desirable properties, in particular wrt the actual treatment effect.

**Strengths:**

* A well written paper that fits well in the benchmark track
* Clearly fills a need in the treatment effects community
* an easy to use dataset as the authors provide a package to include it in other code bases

**Additional Feedback:**

na.

**Documentation:**

I also had a suggestion with respect to this: It would be nice to include a small code sample on how to use the proposed method to use RORCO to evaluate a treatment effects model (such as Appendix A).

**Ethics:**

I believe the authors used a publicly available dataset to build a semi synthetic dataset. Therefore I don't think any additional licensising etc. to be required.

**Limitations:**

na.

**Opportunities For Improvement:**

* There is currently a gap between RealRORCO and RORCO in terms of observed treatment effect. why is this the case? is this a problem?
* Perhaps it would be nice to move some of the theoretical analysis (I believe this is less suited for this track) in favour of a code sample on how to use the dataset using the proposed package.
* is there anyway to _show_ the problems associated with other benchmark datasets beyond the descriptive statistics in table 1? From the introduction:
> the datasets are often repurposed from other tasks and (somewhat crudely) adapted to treatment effect estimation.

What does this mean and how does it manifest when benchmarking methods on such datasets?

**Relation To Prior Work:**

I think my questions revolve somewhat around this. Essentially, I would like to know why other datasets are bad, and this one good (beyond some nice descriptive stats).

**Summary And Contributions:**

The authors propose RORCO, a novel dataset based on natural experiments with the specific aim to evaluate treatment effect models. Given that, on RORCO and other datasets, it seems that doubly-robust methods seem to outperform others, the authors continue and provide a meaningful analysis as to why this may be the case. From their theoretical analysis, the authors propose Double-Double, a novel DR-learner, with double weighting.

---

> ### Author Rebuttal · Authors · 2024-08-16
>
> Dear Reviewer,
>
> Thank you for your time and feedback! We address your specific comments and questions below.
>
> **Different Outcomes in RORCO and RORCO Real**
>
> As you point out, the dataset with real outcomes (RORCO Real) and the synthetic dataset (RORCO) have different outcomes. This is intentional. The RORCO outcomes reflect the suggestions of literacy experts (more details below) while the RORCO Real outcomes are based on a best guess based proximity connection between well-child visits and standardized test scores. We intentionally make the synthetic outcomes reflect the guidance of the literacy experts rather than tailoring to the real (noisy) outcomes we observe. We believe the different outcomes are a benefit, making our benchmark more robust.
>
> The literacy experts we talked to made the following suggestions for the synthetic outcomes:
>
> (1) The outcomes should be inversely related to the propensity. That is, students are more likely to participate in the literacy program if they have lower literacy proficiency.
>
> (2) The treatment effect should be negligible until the propensities reach one half and increasing after. That is, students who are less likely to participate in the program, and so have higher literacy proficiency as per (1), will not benefit from the program because they already have sufficient resources. In contrast, students who are more likely to participate in the program, and so have lower literacy proficiency as per (1), will benefit from the program in proportion to their literacy needs.
>
> Based on (1) and (2), we made the control outcomes vary between 0 and 1 with an inverse linear relationship to propensity scores. We made the treatment outcomes align with the control outcomes until .5 and then increasingly separate (with a negated square root added to the control outcomes).
>
> **Example Code**
>
> We created a Jupyter notebook to demonstrate the functionality of the package (available [here](https://github.com/rtealwitter/naturalexperiments/blob/main/demo.ipynb)) and we included all the code to recreate the paper figures in the repo (available [here](https://github.com/rtealwitter/naturalexperiments/tree/main/paper_experiments)). In order to emphasize the utility of the package, we have added code samples to the main body of the paper. Thank you for the suggestion!
>
> **Qualitative Dataset Evaluation**
>
> Broadly, many of the existing datasets do not solve a real treatment effect problem so tailoring algorithms to these datasets may not lead to better performance in real settings. We’ll expand on what we mean for specific datasets:
>
> * The Twins dataset is based on an observational study of twins. The “treatment” that is commonly used is whether a child was born heavier than their twin. This is not ideal because it doesn’t incorporate the difference in weight: Presumably, if weight will impact the child’s life, then a difference of one ounce is very different from the difference of one pound.
>
> * The News dataset is synthetically generated by computer scientists to mimic a preference for reading on mobile devices without, it seems, any domain expert guidance (details in Section 6.2 [here](https://proceedings.mlr.press/v48/johansson16.pdf)).
>
> * The ACIC 2017 and 2018 datasets were synthetically generated for a competition. They make strong assumptions on the outcomes (details in Section 2 [here](https://arxiv.org/pdf/1905.09515)) and, it seems, also do not include any domain experts in the process.
>
> * The IHDP dataset is based on real covariates (from an observational study starting in 1985) but with simplistic synthetic outcomes. The outcomes are both based on a random linear combination of the covariates (details in Section 4.1 [here](https://www.tandfonline.com/doi/pdf/10.1198/jcgs.2010.08162?casa_token=M7NnlVhJMJkAAAAA:VN5OwzcQb8i4DkEEFEf5wwnBHewH4WrY37aLTyIZuTPmzLOuUDEE987o56_xI7ek_GZ3-5EJu6IovA)): the control outcomes are a unit-normal distribution centered at the linear combination while the control outcomes are a unit-normal distribution centered at the linear combination plus 4. Due to the simplicity of the outcomes, they may not reflect the complexity of real settings.
>
> The Jobs dataset *is* based on a natural experiment on the effect of job training. The control outcomes are incomes from before the training in 1975 while the treatments are incomes from after the training in 1978. Unfortunately, the dataset only has 800 observations and 8 covariates.
>
> Of course, creating a treatment effect dataset is quite challenging and we do not do it perfectly. The RORCO Real dataset has real covariates and outcomes but makes a strong assumption to account for the lack of personally identifiable information. The RORCO dataset has real covariates but synthetic outcomes which *are* designed with domain expert guidance.

---

### Official Review · Reviewer_NCag · 2024-07-25
**Review of "Benchmarking Estimators for Natural Experiments: A Novel Dataset and a Doubly Robust Algorithm"**

**Rating:** 8
**Confidence:** 2

**Review:**

Quality: The research quality of this work is high, with a robust methodological approach and thorough analyses. The authors have implemented an extensive benchmark, contributing to the field of treatment effect estimation, especially in the context of natural experiments. The evaluation of the variation of real-world conditions like sample size, treatment correlation, and propensity score accuracy seems appropriate and exhaustive. The introduction of a new dataset derived from a real-world scenario adds practical relevance and applicability. To fill in the gap of lack of outcomes, the authors additionally propose a semi-synthetic version of the dataset, designed with the support of domain experts. While some assumptions seem fragile, the discussion enriches the work.

Clarity: The paper is generally well-written and presents a good structure, which is easy to follow. Nonetheless, similar to other presented concepts, the reader, especially one not so familiar with the definition, would benefit from including a brief definition of doubly robust estimators. Other specific suggestions will follow in the corresponding Clarity section below.

Originality: While the use of doubly robust estimators and the training split approach are not novel, the specific implementation as a benchmark and the context of a new dataset add originality.

Significance: The paper's significance lies in its practical contributions: a new dataset, an extensive benchmark, and a novel doubly robust estimator. The benchmark, including the other elements, shows potential to stimulate new research and applications in the field of treatment effect estimation based on natural experiments.

**Strengths:**

* Comprehensive and extensible benchmark framework for evaluating treatment effect estimators as the sample size, propensity score accuracy, and correlation vary.
* Introduction of a novel dataset derived from real-world natural experiments, as well as semi-synthetic version designed with the help of domain experts to allow for additional evaluations.
* Theoretical analysis (with proof) resulting in a closed form variance expression and proposal of a new doubly robust estimator.
* Release of the dataset and benchmark as an open-source Python package.

**Additional Feedback:**

N/A

**Clarity:**

The paper is well-written and easy to follow, but could improve by defining technical terms such as doubly robust estimators and by correcting minor typographical errors.

**Correctness:**

The claims of this work appear to be correct based on the methodologies and analyses presented. The dataset is constructed in a sound manner, and even the semi-synthetic version is designed with the support of domain experts, even if there is a strong assumption that might be challenged. The evaluation methods, factors to vary for evaluation, and experiment design seem appropriate and performed correctly.

**Documentation:**

The dataset documentation is detailed and sufficient, covering data collection, description and availability (some details in the manuscript, more on the Github repository). The ethical and responsible use of the dataset is confirmed, and the authors plan to maintain the dataset indefinitely on GitHub.

**Ethics:**

There do not appear to be any significant ethical concerns with the submission, although the checklist's items on specific points like consent and personally identifiable information are answered a bit ambiguously with "All datasets in our experiments are freely available for public use"

**Limitations:**

The authors acknowledge the main limitations, such as the strong assumption of the observational RORCO dataset on clinic-school proximity to determine treatment since individuals cannot be tracked; and the fact that the theoretical framework derived only applies to doubly robust estimators with a training split. If space would allow it, discussing potential biases introduced by the assumptions and the synthetic outcomes would be positive. Considering a net positive impact of their research seems reasonable.

**Opportunities For Improvement:**

Only minor suggestions:
* For improved clarity, the authors should include a brief definition of doubly robust estimators.
* Related work could benefit from adding references on sample splitting or cross-fitting, as related to the training split approach.
* Table 3 would benefit from a separating line between doubly robust estimators that do and do not use data splitting.
* A few typos to correct, e.g., "Depending on whether the observation receives the treatment or control, we observe the treatment outcome
𝑦(1)𝑖 ∈ 𝑅  or the control outcome 𝑦(1)𝑖 ∈ 𝑅, but not both." The control outcome should be 𝑦(0)𝑖, it seems.

**Relation To Prior Work:**

The paper discusses its relation to prior work but could improve by explicitly mentioning the previous use of sample splitting and cross-fitting
 in relation to the training split strategy the authors employ, focusing on how the latter differs from the previous approaches.

**Summary And Contributions:**

This work introduces a new dataset derived from a nonprofit focused on early childhood literacy, intended to evaluate treatment effects using natural experiments. It presents an extensive benchmark of over 20 established estimators, concluding that doubly robust estimators generally outperform other, more complex ones. The authors propose a new doubly robust estimator with a novel loss function and release the dataset and benchmark as a Python package to ease further research. Key contributions include the new dataset, the benchmark framework, the novel doubly robust estimator Double-Double, and the Python package for the community.

---

> ### Author Rebuttal · Authors · 2024-08-16
>
> Dear Reviewer,
>
> Thank you for your time and feedback!
>
> **Changes:**
>
> We have made the changes as per your suggestions for the revised version i.e., a definition of doubly robust estimators, related work on sample splitting, a separating line for clarity in Table 3, a typo correction, and a discussion about the biases from our data assumptions.
>
> **Personally Identifiable Information:**
>
> We would like to provide more clarity about personally identifiable information in the RORCO dataset. There are two datasets that we combine, neither of which contains personally identifiable information. The first dataset, publicly accessible from the Colorado Department of Education website, is aggregated by grade (at least ten students) and only reports mean statistics e.g., average CMAS test score. The second dataset, from Reach Out and Read Colorado, only contains counts of the number of clinic well-child visits in 6 month intervals. The lack of personally identifiable information is what requires us to make a proximity assumption to link well-child visits to grades.

---

> > ### Comment · Reviewer_NCag · 2024-08-29
> >
> > Thank you for considering the changes and comments, although I don't have access to the revised document. I'd like to check mostly the new definition for doubly robust estimators, and the discussion on the biases of the data assumptions. I've also checked the rebuttal to other reviewers and appreciate the effort to improve the paper further.

---

> > ### Author Response · Authors · 2024-08-29
> >
> > Dear Reviewer,
> >
> > Thank you for your comment! As you say, we unfortunately do not have a way to upload the revised version of the paper but we provide the definition of doubly robust estimators and the discussion of biases stemming from the data assumptions below.
> >
> > > I'd like to check mostly the new definition for doubly robust estimators,
> >
> > Recall we have $n$ individuals with covariates $\mathbf{x}_i$, control outcomes $y_i^{(0)}$, and treatment outcomes $y_i^{(1)}$ for $i \in [n]$. We use $\mathbf{z} \in \{0,1\}^n$ to indicate whether each individual received the treatment or control.
> >
> > Definition [Doubly Robust Estimator]
> >
> > Let $f^{(0)}, f^{(1)}: \mathbb{R}^d \to \mathbb{R}$ be learned functions for the control and treatment outcomes, respectively. Let $p_i$ be the (estimated) propensity score for individual $i \in [n]$ with outcomes $\mathbf{x}_i$. A doubly robust estimator is given by
> >
> > $\tau(\mathbf{z}) = \frac1{n} \sum_{i=1}^n \left( \frac{y_i^{(1)}  - f^{(1)}(\mathbf{x}_i)}{p_i} {1}[z_i=1] - \frac{y_i^{(0)} - f^{(0)}(\mathbf{x}_i)}{1-p_i} {1}[z_i \neq 1] + f^{(1)}(\mathbf{x}_i) - f^{(0)}(\mathbf{x}_i) \right).$
> >
> > Different doubly robust estimators can be obtained by changing the way the functions $f^{(0)}$ and $f^{(1)}$ are learned, or including different estimates of the propensity scores.
> >
> > > and the discussion on the biases of the data assumptions.
> >
> > We make several assumptions while building the RORCO and RORCO Real datasets.
> >
> > For the RORCO Real dataset, we must make assumptions in order to determine whether a class (the most granular education data available) received the RORCO “treatment”. These assumptions potentially bias the resulting datasets in the following ways:
> >
> > * By using class (as opposed to individual students) as observations, we potentially reduce the effect that RORCO can have. For example, if only half the class received the RORCO treatment then the effect on their literacy outcomes will be weaker.
> >
> > * By using proximity in rural communities to determine whether classes received the RORCO treatment, we change the distribution of the data to only reflect sparsely populated geographic areas. Further, we make an unverified assumption that students did not move over the course of several years in these rural communities, i.e., they attend school near where they lived as a child.
> >
> > For the RORCO dataset, we must make assumptions to synthetically generate outcomes. We attempt to make the assumptions realistic by relying on literacy experts. The literacy experts we spoke with suggested the following assumptions:
> >
> > (1) The outcomes should be inversely related to the propensity. That is, students are more likely to participate in the literacy program if they have lower literacy proficiency.
> >
> > (2) The treatment effect should be negligible until the propensities reach one half and increasing after. That is, students who are less likely to participate in the program, and so have higher literacy proficiency as per (1), will not benefit from the program because they already have sufficient resources. In contrast, students who are more likely to participate in the program, and so have lower literacy proficiency as per (1), will benefit from the program in proportion to their literacy needs.
> >
> > Based on (1) and (2), we made the control outcomes vary between 0 and 1 with an inverse linear relationship to propensity scores. We made the treatment outcomes align with the control outcomes until .5 and then increasingly separate (with a negated square root added to the control outcomes).
> >
> > While conforming to expert understanding, these assumptions do not necessarily reflect what actually happens in the real world. As a result, fine-tuning estimators *only* on these synthetic outcomes may result in algorithms that are not applicable to real settings.

---

> > > ### Comment · Reviewer_NCag · 2024-09-02
> > >
> > > Thank you for providing the answer to my request in your response. I'm happy with the content, but I have small note: "... with outcomes x_i", should it be the covariates x_i, or the outcomes y_i?

---

> > > > ### Author Response · Authors · 2024-09-02
> > > >
> > > > Good catch! It should be “covariates $x_i$”. We will make sure to do a thorough pass for typos such as this before submitting the revised version.

---

### Decision · Program_Chairs · 2024-09-26

**Decision:**

Accept (Poster)

**Comment:**

This paper introduces a new dataset for studying treatment effect estimation from natural experiments related to childhood literacy. The paper compares 20 estimators, suggesting that doubly robust estimators tend to outperform other estimators. Like many datasets used in causality, this one is used in a semisynthetic fashion, with a synthetic outcome model. The reviewers rated the paper highly on account of the practical contributions. I see no reason to overturn the reviewers judgment.